# Learning Fast and Slow for Online Time Series Forecasting

**Quang Pham**[†][*]**, Chenghao Liu**[§]**, Doyen Sahoo**[§]**, Steven C.H. Hoi** [§,‡]

[§] Salesforce Research Asia

[†] Institute for Infocomm Research (I²R), Agency for Science, Technology and Research (A*STAR),
1 Fusionopolis Way, #21-01, Connexis South Tower, Singapore 138632, Republic of Singapore

[‡] School of Computing and Information Systems, Singapore Management University

## Abstract

Despite the recent success of deep learning for time series forecasting, these methods are not scalable for many real-world applications where data arrives sequentially. Training deep neural forecasters on the fly is notoriously challenging because of their limited ability to adapt to non-stationary environments and remember old knowledge. We argue that the fast adaptation capability of deep neural networks is critical and successful solutions require handling changes to both new and recurring patterns effectively. In this work, inspired by the Complementary Learning Systems (CLS) theory, we propose Fast and Slow learning Network (FSNet) as a novel framework to address the challenges of online forecasting. Particularly, FSNet improves the slowly-learned backbone by dynamically balancing fast adaptation to recent changes and retrieving similar old knowledge. FSNet achieves this mechanism via an interaction between two novel complementary components: (i) a per-layer adapter to support fast learning from individual layers, and (ii) an associative memory to support remembering, updating, and recalling repeating events. Extensive experiments on real and synthetic datasets validate FSNet's efficacy and robustness to both new and recurring patterns. Our code is publicly available at: https://github.com/salesforce/fsnet/.

## 1 Introduction

Time series forecasting plays an important role in both research and industries. Correctly forecast time series can greatly benefit various business sectors such as traffic management and electricity consumption (Hyndman & Athanasopoulos, 2018). As a result, tremendous efforts have been devoted to develop better forecasting models (Petropoulos et al., 2020; Bhatnagar et al., 2021; Triebe et al., 2021), with a recent success of deep neural networks (Li et al., 2019; Xu et al., 2021; Yue et al., 2021; Zhou et al., 2021) thanks to their impressive capabilities to discover hierarchical latent representations and complex dependencies. However, such studies focus on the batch learning setting which requires the whole training dataset to be made available a priori and implies the relationship between the input and outputs remains static throughout. This assumption is restrictive in real-world applications, where data arrives in a stream and the input-output relationship can change over time (Gama et al., 2014). In such cases, re-training the model from scratch could be time consuming. Therefore, it is desirable to train the deep forecaster online (Anava et al., 2013; Liu et al., 2016) using only new samples to capture the changing dynamic of the environment.

Despite the ubiquitous of online learning in many real-world applications, training deep forecasters online remains challenging for two reaons. First, naively train deep neural networks on data streams requires many samples to converge (Sahoo et al., 2018; Aljundi et al., 2019a) because the offline training benefits such as mini-batches or training for multiple epochs are not available. Therefore, when a distribution shift happens (Gama et al., 2014), such cumbersome models would require many samples to learn new concepts with satisfactory results. Overall, deep neural networks, although possess strong representation learning capabilities, lack a mechanism to facilitate successful learning on data streams. Second, time series data often exhibit recurrent patterns where one pattern

---

[*]Part of this work was done during Quang Pham's internship at Salesforce Research Asia. The final version was prepared at I²R, A*STAR. Correspondence to: phamhq@i2r.a-star.edu.sg

could become inactive and re-emerge in the future. Since deep networks suffer from catastrophic forgetting (McCloskey & Cohen, 1989), they cannot retain prior knowledge and result in inefficient learning of recurring patterns, which further hinders the overall performance. Consequently, online time series forecasting with deep models presents a promising yet challenging problem.

To address the above limitations, we radically formulate online time series forecasting as an *online, task-free continual learning* problem (Aljundi et al., 2019a;b). Particularly, continual learning requires balancing two objectives: (i) utilizing past knowledge to facilitate fast learning of current patterns; and (ii) maintaining and updating the already acquired knowledge. These two objectives closely match the aforementioned challenges and are usually referred to as the *stability-plasticity* dilemma (Grossberg, 1982). With this connection, we develop an effective online time series forecasting framework motivated by the *Complementary Learning Systems (CLS) theory* (McClelland et al., 1995; Kumaran et al., 2016), a neuroscience framework for human continual learning. Specifically, the CLS theory suggests that humans can continually learn thanks to the interactions between the *hippocampus* and the *neocortex*, which supports the consolidation, recall, and update such experiences to form a more general representation, which supports generalization to new experiences.

This work develops FSNet (Fast-and-Slow learning Network) to enhance the sample efficiency of deep networks when dealing with distribution shifts or recurring concepts in online time series forecasting. FSNet's key idea for fast learning is its ability to always improve the learning at current steps instead of explicitly detecting changes in the environment. To do so, FSNet employs a per-layer adapter to model the temporal consistency in time series and adjust each intermediate layer to learn better, which in turn improve the learning of the whole deep network. In addition, FSNet further equip each adapter with an associative memory (Kaiser et al., 2017) to store important, recurring patterns observed. When encountering such events, the adapter interacts with its memory to retrieve and update the previous actions to further facilitate fast learning. Consequently, the adapter can model the temporal smoothness in time series to facilitate learning while its interactions with the associative memories support remembering and improving the learning of recurring patterns.

In summary, our work makes the following contributions. First, we radically formulate learning fast in online time series forecasting with deep models as a continual learning problem. Second, motivated by the CLS theory, we propose a fast-and-slow learning paradigm of FSNet to handle both the fast changing and long-term knowledge in time series. Lastly, we conduct extensive experiments with both real and synthetic datasets to demonstrate FSNet's efficacy and robustness.

## 2 PRELIMINARY AND RELATED WORK

This section provides the necessary background of time series forecasting and continual learning.

### 2.1 TIME SERIES FORECASTING SETTINGS

Let $\mathcal{X} = (\boldsymbol{x}_1, \ldots, \boldsymbol{x}_T) \in \mathbb{R}^{T \times n}$ be a time series of $T$ observations, each has $n$ dimensions. The goal of time series forecasting is that given a look-back window of length $e$, ending at time $i$: $\mathcal{X}_{i,e} = (\boldsymbol{x}_{i-e+1}, \ldots, \boldsymbol{x}_i)$, predict the next $H$ steps of the time series as $f_\omega(\mathcal{X}_{i,H}) = (\boldsymbol{x}_{i+1}, \ldots, \boldsymbol{x}_{i+H})$, where $\omega$ denotes the parameter of the forecasting model. We refer to a pair of look-back and forecast windows as a sample. For multiple-step forecasting ($H > 1$) we follow the standard approach of employing a linear regressor to forecast all $H$ steps in the horizon simultaneously (Zhou et al., 2021).

**Online Time Series Forecasting** is ubiquitous is many real-world scenarios (Anava et al., 2013; Liu et al., 2016; Gultekin & Paisley, 2018; Aydore et al., 2019) due to the sequential nature of data. In this setting, there is no separation of training and evaluation. Instead, learning occurs over a sequence of rounds. At each round, the model receives a look-back window and predicts the forecast window. Then, the true answer is revealed to improve the model's predictions of the incoming rounds (Hazan, 2019). The model is commonly evaluated by its accumulated errors throughout learning (Sahoo et al., 2018). Due to its challenging nature, online time series forecasting exhibits several challenging sub-problems, ranging from learning under concept drifts (Gama et al., 2014), to dealing with missing values because of the irregularly-sampled data (Li & Marlin, 2020; Gupta et al., 2021). In this work, we focus on the problem of fast learning (in terms of sample efficiency) under concept drifts by improving the deep network's architecture and recalling relevant past knowledge. There is also a rich literature of Bayesian continual learning to address regression problems (Smola et al., 2003; Kurle et al., 2019; Gupta et al., 2021). However, such formulation follow the Bayesian

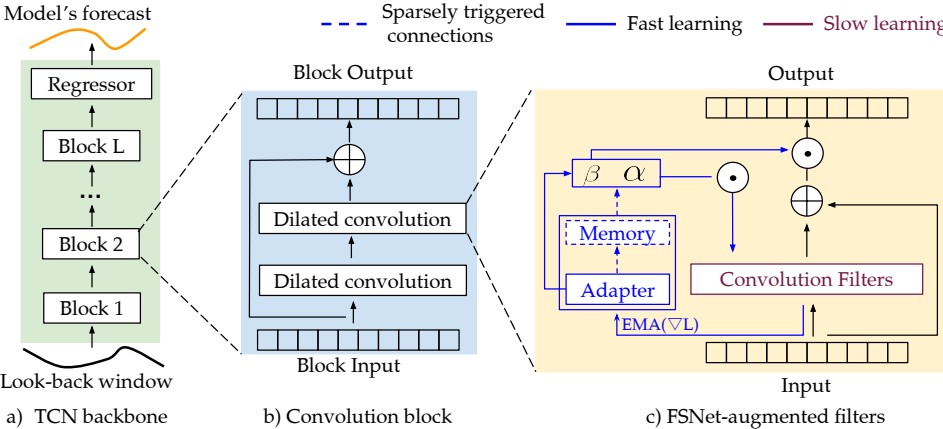

a) TCN backbone      b) Convolution block      c) FSNet-augmented filters

Figure 1: An overview of FSNet. A standard TCN backbone (a) of L dilated convolution stacks (b). Each convolution filter in FSNet is equipped with an adapter and associative memory to facilitate fast learning by monitoring the backbone's gradient EMA. Best viewed in colors.

framework, which allows for forgetting of past knowledge and does not have an explicit mechanism for fast learning (Huszár, 2017; Kirkpatrick et al., 2018). Moreover, such studies did not focus on deep neural networks and it is non-trivial to extend to the setting of our study.

## 2.2 Continual Learning

Continual learning (Kirkpatrick et al., 2017) is an emerging topic aiming to build intelligent agents that can learn a series of tasks sequentially, with only limited access to past experiences. A continual learner must achieve a good trade-off between maintaining the acquired knowledge of previous tasks and facilitating the learning of future tasks, which is also known as the *stability-plasticity* dilemma (Grossberg, 1982; 2013). Due to its connections to humans' learning, several neuroscience frameworks have motivated the development of various continual learning algorithms. One popular framework is the complementary learning systems theory for a dual learning system (McClelland et al., 1995; Kumaran et al., 2016). Continual learning methods inspired from the CLS theory augments the slow, deep networks with the ability to quickly learn on data streams, either via the experience replay mechanism (Lin, 1992; Riemer et al., 2019; Rolnick et al., 2019; Aljundi et al., 2019a; Buzzega et al., 2020) or via explicit modeling of each of the fast and slow learning components (Pham et al., 2021a; Arani et al., 2021). Such methods have demonstrated promising results on controlled vision or language benchmarks. In contrast, our work addresses the online time series forecasting challenges by formulating them as a continual learning problem.

## 3 Proposed Framework

This section formulates the online time series forecasting as a task-free online continual learning problem and details the proposed FSNet framework.

### 3.1 Online Time Series Forecasting as a Continual Learning Problem

Our formulation is motivated by the locally stationary stochastic processes observation, where a time series can be split into a sequence of stationary segments (Vogt, 2012; Dahlhaus, 2012; Das & Nason, 2016). Since the same underlying process generates samples from a stationary segment, we refer to forecasting each stationary segment as a learning task for continual learning. We note that this formulation is general and encompasses existing learning paradigms. For example, splitting into only one segment indicates no concept drifts, and learning reduces to online learning in stationary environments (Hazan, 2019). Online continual learning (Aljundi et al., 2019a) corresponds to the case of there are at least two segments. Moreover, we also do not assume that the points of task switch are given to the model, which is a common setting in many continual learning studies (Kirkpatrick et al., 2017; Lopez-Paz & Ranzato, 2017). Manually obtaining such information in real-world data can be expensive because of the missing or irregularly sampled data (Li & Mar-

lin, 2020; Farnoosh et al., 2021). Therefore, our formulation corresponds to the online, task-free continual learning formulation (Aljundi et al., 2019a;b; Hu et al., 2020; Cai et al., 2021).

We now discuss the differences between our formulation with existing studies. First, most existing task-free continual learning frameworks (Aljundi et al., 2019b; Pham et al., 2021a) are developed for image data, which vastly differs from time series. The input and label spaces of images are different (continuous vs discrete) while time series' input and output share the same real-valued space. Additionally, the image's label changes significantly across tasks while time series data changes gradually over time with no clear boundary. Moreover, time series exhibits strong temporal information among consecutive samples, which does not exist in image data. Therefore, it is non-trivial to simply apply existing continual learning methods to time series and successful solutions requires carefully handling unique characteristics from time series data.

Second, time series evolves and old patterns may not reappear exactly in the future. Thus, we are not interested in remembering old patterns precisely but predicting **how they will evolve**. *For example, we do not need to predict the electricity consumption over the last winter. But it is more important to predict the electricity consumption this winter, assuming that it is likely to have a similar pattern as the last one.* Therefore, we do not need a separate test set for evaluation, but training follows the online learning setting where a model is evaluated by its accumulated errors throughout learning.

## 3.2 FAST AND SLOW LEARNING NETWORKS (FSNET)

FSNet always leverages past knowledge to improve the learning in the future (Section 3.2.1), which is akin to facilitating forward transfer in continual learning (Lopez-Paz & Ranzato, 2017). Additionally, FSNet remembers repeating events and continue to learn them when they reappear (Section 3.2.2, which is akin to preventing catastrophic forgetting (Kirkpatrick et al., 2017). We consider Temporal Convolutional Network (TCN) (Bai et al., 2018) as the backbone deep neural network to extract a time series feature representation due to the simple forward architecture and promising results (Yue et al., 2021). The backbone has $L$ layer with parameters $\boldsymbol{\theta} = \{\boldsymbol{\theta}_l\}_{l=1}^L$. FSNet improves the TCN backbone with *two* complementary components: a per-layer adapter $\phi_l$ and a per-layer associative memory $\mathcal{M}_l$. Thus, the total *trainable* parameters is $\boldsymbol{\omega} = \{\boldsymbol{\theta}_l, \phi_l\}_{l=1}^L$ and the total associative memory is $\mathcal{M} = \{\mathcal{M}_l\}_{l=1}^L$. We also use $\boldsymbol{h}_l$ and $\tilde{\boldsymbol{h}}_l$ to denote the original feature and adapter feature map of the $l-$layer. Figure 1 provides an illustration of FSNet.

### 3.2.1 FAST LEARNING MECHANISM

The key observation allowing for a fast learning is to facilitate the learning of each intermediate layer via the following observation: *the partial derivative $\nabla_{\boldsymbol{\theta}_l}\ell$ characterizes the contribution of layer $\boldsymbol{\theta}_l$ to the forecasting loss $\ell$.* Traditional training schemes simply move the parameters along this gradient direction, which results in ineffective online learning (Sahoo et al., 2018; Phuong & Lampert, 2019). Moreover, time series data exhibits strong temporal consistency across consecutive samples, which is not captured by existing training frameworks. Putting these observations together, we argue that an exponential moving average (EMA) of the partial derivative can provide meaningful information about the temporal smoothness in time series. Consequently, leveraging this knowledge can improve the learning of each layer, which in turn improves the whole network's performance.

To utilize the gradient EMA, we propose to treat it as a context to support fast learning via the feature-wise transformation framework (Perez et al., 2018; Dumoulin et al., 2018; Pham et al., 2021b; Yin et al., 2021). Particularly, we propose to equip each layer with an adapter to map the layer's gradient EMA to a set of smaller, more compact transformation coefficients. These coefficients are applied on the corresponding layer's parameters and feature so that they can leverage the temporal consistency to learn better. We first define the EMA of the $l-$layer's partial derivative as:

$$\hat{\boldsymbol{g}}_l \leftarrow \gamma\hat{\boldsymbol{g}}_l + (1-\gamma)\boldsymbol{g}_l^t, \tag{1}$$

where $\boldsymbol{g}_l^t$ denotes the gradient of the $l-$th layer at time $t$ and $\hat{\boldsymbol{g}}_l$ denotes the EMA. The adapter takes $\hat{\boldsymbol{g}}_l$ as input and maps it to the adaptation coefficients $\boldsymbol{u}_l$. Then, an adapter for the $l-$th layer is a linear layer that maps the context $\hat{\boldsymbol{g}}_l$ to a set of transformation coefficients $\boldsymbol{u}_l = [\boldsymbol{\alpha}_l; \boldsymbol{\beta}_l]$. In this work, we consider a two-stage transformations (Yin et al., 2021) which involve a weight and bias transformation coefficients $\boldsymbol{\alpha}_l$ and a feature transformation coefficients $\boldsymbol{\beta}_l$.

The adaptation process for a layer $\boldsymbol{\theta}_l$ is summarized as:

$$[\boldsymbol{\alpha}_l, \boldsymbol{\beta}_l] = \boldsymbol{u}_l, \text{ where } \boldsymbol{u}_l = \boldsymbol{\Omega}(\hat{\boldsymbol{g}}_l; \boldsymbol{\phi}_l) \tag{2}$$

$$\text{Weight adaptation: } \tilde{\boldsymbol{\theta}}_l = \text{tile}(\boldsymbol{\alpha}_l) \odot \boldsymbol{\theta}_l, \text{ and} \tag{3}$$

$$\text{Feature adaptation: } \tilde{\boldsymbol{h}}_l = \text{tile}(\boldsymbol{\beta}_l) \odot \boldsymbol{h}_l, \text{ where } \boldsymbol{h}_l = \tilde{\boldsymbol{\theta}}_l \circledast \tilde{\boldsymbol{h}}_{l-1}. \tag{4}$$

Here, $\boldsymbol{h}_l$ is a stack of $I$ features maps with $C$ channels and length $Z$, $\tilde{\boldsymbol{h}}_l$ is the adapted feature, $\tilde{\boldsymbol{\theta}}_l$ denotes the adapted weight, $\odot$ denotes the element-wise multiplication, and $\text{tile}(\boldsymbol{\alpha}_l)$ denotes that the weight adaptor is applied per-channel on all filters via a tile function that repeats a vector along the new axes. A naive implementation of Equation 2 directly maps the model's gradient to the adaptation coefficients and results in a very high dimensional mapping. Therefore, we implement the *chunking* operation (Ha et al., 2016) to split the gradient into equal size chunks and then maps each chunk to an element of the adaptation coefficients. We denote this chunking operator as $\Omega(\cdot; \boldsymbol{\phi}_l)$ and provide the detailed description in Appendix C.

### 3.2.2 REMEMBERING RECURRING EVENTS WITH AN ASSOCIATIVE MEMORY

In time series, old patterns may reappear and it is imperative to leverage our past actions to improve the learning outcomes. In FSNet, an adaptation to a pattern is represented by the coefficients $\boldsymbol{u}$, which we argue to be useful to learn repeating events. Specifically, $\boldsymbol{u}$ represents how we adapted to a particular pattern in the past; thus, storing and retrieving the appropriate $\boldsymbol{u}$ may facilitate learning the corresponding pattern when they reappear. Therefore, as the second key element in FSNet, we implement an associative memory to store the adaptation coefficients of repeating events encountered during learning. In summary, beside the adapter, we equip each layer with an additional associative memory $\mathcal{M}_l \in \mathbb{R}^{N \times d}$ where $d$ denotes the dimension of $\boldsymbol{u}_l$, and $N$ denotes the number of elements, which we fix as $N = 32$ by default.

**Sparse Adapter-Memory Interactions** Interacting with the memory at every step is expensive and susceptible to noises. Thus, we propose to trigger this interaction subject to a substantial representation change. Interference between the current and past representations can be characterized in terms of a dot product between the gradients (Lopez-Paz & Ranzato, 2017; Riemer et al., 2019). To this end, together with the gradient EMA in Equation 2, we deploy another gradient EMA $\hat{\boldsymbol{g}}'_l$ with a smaller coefficient $\gamma' < \gamma$ and measure their cosine similarity to trigger the memory interaction as:

$$\text{Trigger if}: \cos(\hat{\boldsymbol{g}}_l, \hat{\boldsymbol{g}}'_l) = \frac{\hat{\boldsymbol{g}}_l \cdot \hat{\boldsymbol{g}}'_l}{||\hat{\boldsymbol{g}}_l|| \, ||\hat{\boldsymbol{g}}_l||} < -\tau, \tag{5}$$

where $\tau > 0$ is a hyper-parameter determining the significant degree of interference. Moreover, we want to set $\tau$ to a relatively high value (e.g. 0.75) so that the memory only remembers significant changing patterns, which could be important and may reappear.

**The Adapter-Memory Interacting Mechanism** Since the current adaptation coefficients may not capture the whole event, which could span over a few samples, we perform the memory read and write operations using the adaptation coefficients's EMA (with coefficient $\gamma'$) to fully capture the current pattern. The EMA of $\boldsymbol{u}_l$ is calculated in the same manner as Equation 1. When a memory interaction is triggered, the adapter queries and retrieves the most similar transformations in the past via an attention read operation, which is a weighted sum over the memory items:

1. Attention calculation: $\boldsymbol{r}_l = \text{softmax}(\mathcal{M}_l \hat{\boldsymbol{u}}_l)$;
2. Top-k selection: $\boldsymbol{r}_l^{(k)} = \text{TopK}(\boldsymbol{r}_l)$;
3. Retrieval: $\tilde{\boldsymbol{u}}_l = \sum_{i=1}^{K} \boldsymbol{r}_l^{(k)}[i] \mathcal{M}_l[i]$,

where $\boldsymbol{r}^{(k)}[i]$ denotes the $i$-th element of $\boldsymbol{r}_l^{(k)}$ and $\mathcal{M}_l[i]$ denotes the $i$-th row of $\mathcal{M}_l$. Since the memory could store conflicting patterns, we employ a sparse attention by retrieving the top-k most relevant memory items, which we fix as $k = 2$. The retrieved adaptation coefficient characterizes how the model reacted to the current pattern in the past, which can improve learning at the present by combining with the current parameters as $\boldsymbol{u}_l \leftarrow \tau \boldsymbol{u}_l + (1 - \tau)\tilde{\boldsymbol{u}}_t$, where we use the same value of $\tau$ as in Equation 5. Then we perform a write operation to update the knowledge stored in $\mathcal{M}_l$ as:

$$\mathcal{M}_l \leftarrow \tau \mathcal{M}_l + (1 - \tau)\hat{\boldsymbol{u}}_l \otimes \boldsymbol{r}_l^{(k)} \text{ and } \mathcal{M}_l \leftarrow \frac{\mathcal{M}_l}{\max(1, ||\mathcal{M}_l||_2)}, \tag{6}$$

where $\otimes$ denotes the outer-product operator, which allows us to efficiently write the new knowledge to the most relevant locations indicated by $\boldsymbol{r}_l^{(k)}$ (Rae et al., 2016; Kaiser et al., 2017). The memory is then normalized to avoid its values exploding. We provide FSNet's pseudo code in Appendix C.2.

## 4 EXPERIMENT

Our experiments aim at investigating the following hypotheses: (i) FSNet facilitates faster adaptation to both new and recurring concepts compared to existing strategies with deep models; (ii) FSNet achieves faster and better convergence than other methods; and (iii) modeling the partial derivative is the key ingredients for fast adaptation. Due to space constraints, we provide the key information of the experimental setting in the main paper and provide full details, including memory analyses, additional visualizations and results in the Appendix.

### 4.1 EXPERIMENTAL SETTINGS

**Datasets** We explore a wide range of time series forecasting datasets. **ETT**[1] (Zhou et al., 2021) records the target value of "oil temperature" and 6 power load features over a period of two years. We consider the ETTh2 and ETTm1 benchmarks where the observations are recorded hourly and in 15-minutes intervals respectively. **ECL** (Electricty Consuming Load)[2] collects the electricity consumption of 321 clients from 2012 to 2014. **Traffic**[3] records the road occupancy rates at San Francisco Bay area freeways. **Weather**[4] records 11 climate features from nearly 1,600 locations in the U.S in an hour intervals from 2010 to 2013.

We also construct two synthetic datasets to explicitly test the model's ability to deal with new and recurring concept drifts. We synthesize a task by sampling $1,000$ samples from a first-order autoregressive process with coefficient $\varphi$: $\text{AR}_\varphi(1)$, where different tasks correspond to different $\varphi$ values. The first synthetic data, **S-Abrupt (S-A)**, contains abrupt, and recurrent concepts where the samples abruptly switch from one AR process to another by the following order: $\text{AR}_{0.1}(1)$, $\text{AR}_{0.4}(1)$, $\text{AR}_{0.6}(1)$, $\text{AR}_{0.1}(1)$, $\text{AR}_{0.3}(1)$, $\text{AR}_{0.6}(1)$. The second data, **S-Gradual (S-G)** contains gradual, incremental shifts, where the shift starts at the last 20% of each task. In this scenario, the last 20% samples of a task is an averaged of two AR process with the order as above. Note that we randomly chose the values of $\varphi$ so that these datasets do not give unfair advantages to any methods.

**Baselines** We consider a suite of baselines from continual learning, time series forecasting, and online learning. First, the **OnlineTCN** strategy that simply trains continuously (Zinkevich, 2003). Second, we consider the Experience Replay (**ER**) (Lin, 1992; Chaudhry et al., 2019) strategy where a buffer is employed to store previous samples and interleave them during the learning of newer ones. We also include three recent advanced variants of ER. First, **TFCL** (Aljundi et al., 2019b) introduces a task-boundaries detection mechanism and a knowledge consolidation strategy by regularizing the networks' outputs (Aljundi et al., 2018). Second, **MIR** (Aljundi et al., 2019a) replace the random sampling in ER by selecting samples that cause the most forgetting. Lastly, **DER++** (Buzzega et al., 2020) augments the standard ER with a knowledge distillation strategy (Hinton et al., 2015). We emphasize that ER and and its variants are strong baselines in the online setting since they enjoy the benefits of training on mini-batches, which greatly reduce noises from singe samples and offer faster, better convergence (Bottou et al., 1998). While the aforementioned baselines use a TCN backbone, we also include **Informer** (Zhou et al., 2021), a recent time series forecasting method based on the transformer architecture (Vaswani et al., 2017). We remind the readers that online time series forecasting have not been widely studied with deep models, therefore, we include general strategies from related fields that we inspired from. Such baselines are competitive and yet general enough to extend to our problem.

**Implementation Details** We split the data into warm-up and online training phases by the ratio of 25:75. We follow the optimization details in (Zhou et al., 2021) by optimizing the $\ell_2$ (MSE) loss with the AdamW optimizer (Loshchilov & Hutter, 2017). In the warm-up phase, we calculate the statistics to normalize online training samples, perform hyper-parameter cross-validation, and pre-

---

[1] https://github.com/zhouhaoyi/ETDataset
[2] https://archive.ics.uci.edu/ml/datasets/ElectricityLoadDiagrams20112014
[3] https://pems.dot.ca.gov/
[4] https://www.ncei.noaa.gov/data/local-climatological-data/

Table 1: Final cumulative MSE and MAE of different methods, "-" indicates the model did not converge. S-A: S-Abrupt, S-G: S-Gradual. Best results are in bold.

| Method | | FSNet | | DER++ | | MIR | | ER | | TFCL | | OnlineTCN | | Informer | |
|---|---|---|---|---|---|---|---|---|---|---|---|---|---|---|---|
| | H | MSE | MAE | MSE | MAE | MSE | MAE | MSE | MAE | MSE | MAE | MSE | MAE | MSE | MAE |
| ETTh2 | 1 | **0.466** | **0.368** | 0.508 | 0.375 | 0.486 | 0.410 | 0.508 | 0.376 | 0.557 | 0.472 | 0.502 | 0.436 | 7.571 | 0.850 |
| ETTh2 | 24 | **0.687** | **0.467** | 0.828 | 0.540 | 0.812 | 0.541 | 0.808 | 0.543 | 0.846 | 0.548 | 0.830 | 0.547 | 4.629 | 0.668 |
| ETTh2 | 48 | **0.846** | **0.515** | 1.157 | 0.577 | 1.103 | 0.565 | 1.136 | 0.571 | 1.208 | 0.592 | 1.183 | 0.589 | 5.692 | 0.752 |
| ETTm1 | 1 | 0.105 | 0.188 | **0.098** | **0.183** | 0.099 | 0.184 | 0.099 | 0.184 | 0.099 | 0.185 | 0.109 | 0.204 | 0.456 | 0.392 |
| ETTm1 | 24 | **0.136** | **0.248** | 0.239 | 0.329 | 0.242 | 0.335 | 0.259 | 0.346 | 0.239 | 0.335 | 0.272 | 0.361 | 0.827 | 0.551 |
| ETTm1 | 48 | **0.129** | **0.245** | 0.264 | 0.355 | 0.271 | 0.362 | 0.288 | 0.372 | 0.242 | 0.344 | 0.280 | 0.371 | 0.853 | 0.533 |
| ECL | 1 | 3.143 | 0.472 | **2.657** | **0.421** | 2.575 | 0.504 | 2.579 | 0.506 | 2.732 | 0.524 | 3.309 | 0.635 | - | - |
| ECL | 24 | **6.051** | **0.997** | 8.996 | 1.035 | 9.265 | 1.066 | 9.327 | 1.057 | 12.094 | 1.256 | 11.339 | 1.196 | - | - |
| ECL | 48 | **7.034** | **1.061** | 9.009 | 1.048 | 9.411 | 1.079 | 9.685 | 1.074 | 12.110 | 1.303 | 11.534 | 1.235 | - | - |
| Traffic | 1 | **0.288** | **0.253** | 0.289 | 0.248 | 0.290 | 0.251 | 0.291 | 0.252 | 0.323 | 0.273 | 0.315 | 0.283 | 0.795 | 0.507 |
| Traffic | 24 | **0.362** | **0.288** | 0.387 | 0.295 | 0.391 | 0.302 | 0.391 | 0.302 | 0.553 | 0.383 | 0.452 | 0.363 | 1.267 | 0.750 |
| WTH | 1 | **0.162** | **0.216** | 0.174 | 0.235 | 0.179 | 0.244 | 0.180 | 0.244 | 0.177 | 0.240 | 0.206 | 0.276 | 0.426 | 0.458 |
| WTH | 24 | **0.188** | **0.276** | 0.287 | 0.351 | 0.291 | 0.355 | 0.293 | 0.356 | 0.301 | 0.363 | 0.308 | 0.367 | 0.370 | 0.417 |
| WTH | 48 | **0.223** | **0.301** | 0.294 | 0.359 | 0.297 | 0.361 | 0.297 | 0.363 | 0.323 | 0.382 | 0.302 | 0.362 | 0.367 | 0.419 |
| S-A | 1 | **1.391** | **0.929** | 2.334 | 1.181 | 2.482 | 1.213 | 2.372 | 1.157 | 2.321 | 1.144 | 2.668 | 1.216 | 3.690 | 1.410 |
| S-A | 24 | **1.299** | **0.904** | 3.598 | 1.439 | 3.662 | 1.450 | 3.375 | 1.360 | 3.415 | 1.366 | 3.904 | 1.491 | 3.657 | 1.426 |
| S-G | 1 | **1.760** | **1.038** | 2.335 | 1.181 | 2.482 | 1.213 | 2.476 | 1.212 | 2.428 | 1.199 | 2.927 | 1.304 | 4.024 | 1.501 |
| S-G | 24 | **1.299** | **0.904** | 3.598 | 1.439 | 3.662 | 1.450 | 3.667 | 1.489 | 3.829 | 1.479 | 3.904 | 1.491 | 3.657 | 1.426 |

Figure 2: Evolution of the cumulative MSE loss during training with forecasting window $H = 24$. In Figure 2.f, each color region denotes a data generating distribution. Best viewed in color.

train the models for the few methods. During online learning, both the epoch and batch size are set to **one** to follow the online learning setting. These configurations are applied to all baselines. We implement a fair comparison by making sure that all baselines use the same total memory budget as our FSNet, which includes *three-times* the network sizes: one working model and two EMA of its gradient. Thus, we set the buffer size of ER, MIR, and DER++ to meet this budget while increasing the backbone size of the remaining baselines. Lastly, for all benchmarks, we set the look-back window length to be 60 and vary the forecast horizon as $H \in \{1, 24, 48\}$.

### 4.2 ONLINE FORECASTING RESULTS

**Cumulative Performance** Table 1 reports the cumulative mean-squared errors (MSE) and mean-absolute errors (MAE) of deep models (TCN and Informer) at the end of training. The reported num-

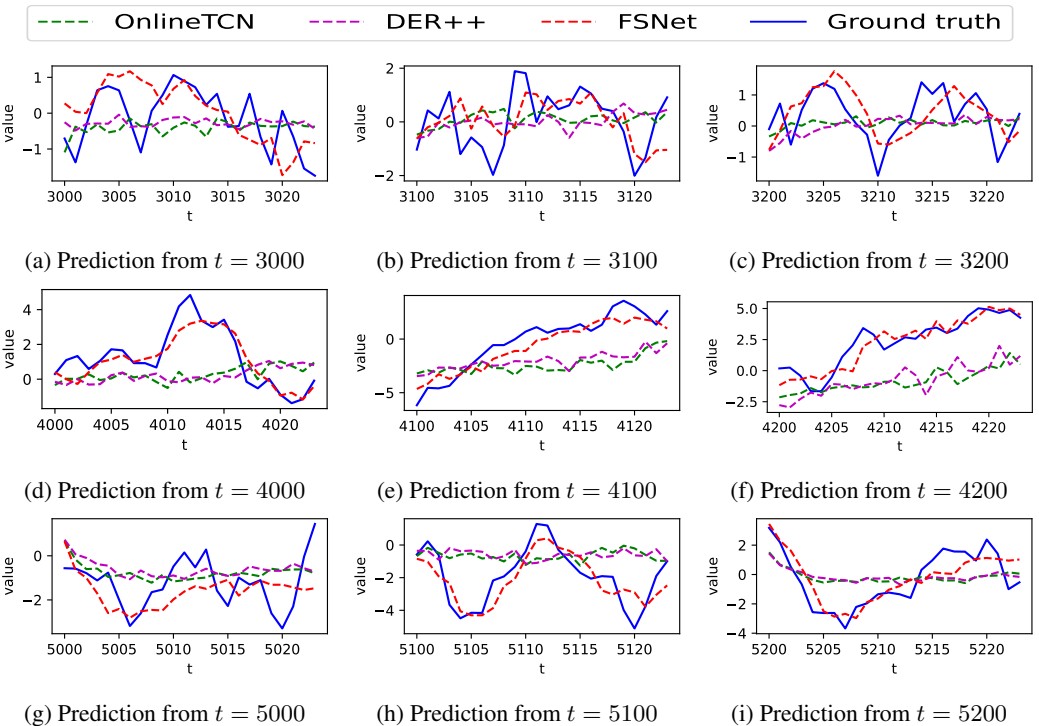

Figure 3: Visualization of the model's prediction throughout the online learning process. We focus on a short horizon of 200 time steps after a concept drift, which is critical for fast learning.

bers are averaged over five runs, and we provide the standard deviations in Table 3, Appendix E.1. We observe that ER and its variants (MIR, DER++) are strong competitors and can significantly improve over the simple TCN strategies. However, such methods still cannot work well under multiple task switches (S-Abrupt). Moreover, no clear task boundaries (S-Gradual) presents an even more challenging problem and increases most models' errors. In addition, previous work has observed that TCN can outperform Informer in the standard time series forecasting (Woo et al., 2022). Here we also observe similar results that Informer does not perform well in the online setting, and is outperformed by other baselines. On the other hand, our FSNet shows promising results on all datasets and outperforms most competing baselines across different forecasting horizons. Moreover, the significant improvements on the synthetic datasets indicate FSNet's ability to quickly adapt to the non-stationary environment and recall previous knowledge, even without clear task boundaries.

**Convergent behaviors of Different Learning Strategies** Figure 2 reports the convergent behaviors on the considered methods. We omit the S-Gradual dataset for spaces because we observe the same behavior as S-Abrupt. Interestingly, we observe that concept drifts are likely to happened in most datasets because of the loss curves' sharp peaks. Moreover, such drifts appear at the early stage of learning, mostly in the first 40% of data, while the remaining half of data are quite stationary. This result shows that the traditional batch training is often too optimistic by only testing the model on the last data segment. The results clearly show the benefits of ER by offering faster convergence during learning compared to OnlineTCN. However, storing the original data may not be applicable in many applications. On S-Abrupt, most baselines demonstrate the inability to quickly recover from concept drifts, indicated by the increasing trend in the error curves. We also observe promising results of FSNet on most datasets, with significant improvements over the baselines on the ETT, WTH, and S-Abrupt datasets. The remaining datasets are more challenging with missing values (Li et al., 2019) and large magnitude varying *within and across* dimensions, which may require calculating better data normalization statistics. While FSNet achieved encouraging results, handling the above challenges can further improve its performance. Overall, the results shed light on the challenges of online time series forecasting and demonstrate promising results of FSNet.

**Visualization** We explore the model's prediction quality on the S-Abrupt since it is a univariate time series. The remaining multivariate real-world datasets are more challenging to visualize. Par-

Table 2: Final comulative MSE and MAE of different FSNet variants. Best results are in bold.

| Method | | FSNet | | | | Variant | | | |
| --- | --- | --- | --- | --- | --- | --- | --- | --- | --- |
| | | M=128 (large) | | M=32 (original) | | No Memory | | Naive | |
| Data | H | MSE | MAE | MSE | MAE | MSE | MAE | MSE | MAE |
| ETTh2 | 24 | **0.616** | **0.456** | 0.687 | 0.467 | 0.689 | 0.468 | 0.860 | 0.555 |
| | 48 | **0.846** | **0.513** | **0.846** | 0.515 | 0.924 | 0.526 | 0.973 | 0.570 |
| Traffic | 1 | **0.285** | **0.251** | 0.288 | 0.253 | 0.294 | 0.252 | 0.330 | 0.282 |
| | 24 | 0.358 | 0.285 | 0.362 | 0.288 | **0.355** | **0.284** | 0.463 | 0.362 |
| S-A | 1 | **1.388** | **0.928** | 1.391 | 0.929 | 1.734 | 1.024 | 3.318 | 1.416 |
| | 24 | **1.213** | **0.870** | 1.299 | 0.904 | 1.390 | 0.933 | 3.727 | 1.467 |
| S-G | 1 | **1.758** | 1.040 | 1.760 | **1.038** | 1.734 | 1.024 | 3.318 | 1.414 |
| | 24 | **1.293** | **0.902** | 1.299 | 0.904 | 1.415 | 0.940 | 3.748 | 1.478 |

ticularly, we are interested in the models' behaviour when an old task's reappear. Therefore, in Figure 3, we plot the model's forecasting at various time points after $t = 3000$. We can see the difficulties of training deep neural networks online in that the model struggles to learn at the early stages, where it only observed a few samples. We focus on the early stages of task switches (e.g. the first 200 samples), which requires the model to quickly adapt to the distribution shifts. With the limited samples per task and the presence of multiple concept drifts, the standard online optimization collapsed to a naive solution of predicting random noises around zero. However, FSNet can successfully capture the time series' patterns and provide better forecasts as learning progresses. Overall, we can clearly see FSNet can provide better quality forecasts compared to other baselines.

### 4.3 ABLATION STUDIES OF FSNET'S DESIGN

This experiment analyzes the contribution of each FSNet's component. First, we explore the benefits of using the associative memory (Section 3.2.2) by constructing a *No Memory* variant that only uses an adapter, without the memory. Second, we further remove the adapter, which results in the *Naive* variant that directly trains the adaptation coefficients $u$ jointly with the backbone. The Naive variant demonstrates the benefits of monitoring the layer's gradients, our key idea for fast adaptation (Section 3.2.1). Lastly, we explore FSNet's scalability by increasing the associative memory size from 32 items (original) to a larger scale of 128 items.

We report the results in Table 2. We first observe that FSNet achieves similar results with the No Memory variant on the Traffic and S-Gradual datasets. One possible reason is the insignificant representation interference in the Traffic dataset and the slowly changing representations in the S-Gradual dataset. In such cases, the representation changes can be easily captured by the adapter alone and may not trigger the memory interactions. In contrast, on ETTh2 and S-Abrupt, which may have sudden drifts, we clearly observe the benefits of storing and recalling the model's past action to facilitate learning of repeating events. Second, the Naive variant does not achieve satisfactory results, indicating the benefits of modeling the temporal smoothness in time series via the use of gradient EMA. Lastly, the large memory variant of FSNet provides improvements in most cases, indicating FSNet's scalability with more budget. Overall, these results demonstrated the complementary of each FSNet's components to deal with different types of concept drift in time series.

## 5 CONCLUSION

We have investigated the potentials and limitations of training deep neural networks for online time series forecasting in non-stationary environments, where they lack the capability to adapt to new or recurring patterns quickly. We then propose Fast and Slow learning Networks (FSNet) by extending the CLS theory for continual learning to online time series forecasting. FSNet augments a neural network backbone with two key components: (i) an adapter for fast learning; and (ii) an associative memory to handle recurrent patterns. Moreover, the adapter sparsely interacts with its memory to store, update, and retrieve important recurring patterns to facilitate learning of such events in the future. Extensive experiments demonstrate the FSNet's capability to deal with various types of concept drifts to achieve promising results in both real-world and synthetic time series data.

ETHIC STATEMENT

In this work, we used the publicly available datasets for experiments. We did not collect human or animal data during this study. Due to the abstract nature of this work, our method does not raise concerns regarding social/gender biases or privacy.

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

## A  ORGANIZATION

This Appendix is organized as follow. First, we provide a more detailed discussion on time series forecasting and continual learning in Appendix B. Then, in Appendix C, we detail the implementation of FSNet. Appendix D discuss the experiments' details, including the synthetic data generation, the loss function, and baselines' descriptions. Lastly, Appendix E provides additional experiment results, including standard deviations, complexity analyses, additional experiments, and visualizations.

# B  EXTENDED RELATED WORK

This section is an extended version of Section 2 where we discuss in more details the existing time series forecasting and continual learning studies.

## B.1  TIME SERIES FORECASTING

Time series forecasting is an important problem and has been extensive studied in the literature. Traditional methods such as ARMA, ARIMA (Box et al., 2015), and the Holt-Winters seasonal methods (Holt, 2004) enjoy theoretical guarantees. However, they lack the model capacity to model more complex interactions of real-world data. As a result, they cannot handle the complex interactions among different dimensions of time series, and often achieve inferior performances compared to deep neural networks on multivariate time series data (Zhou et al., 2021; Oreshkin et al., 2019).

Recently, learning a good time series representation has shown promising results, and deep learning models have surpassed such traditional methods on large scale benchmarks (Rubanova et al., 2019; Zhou et al., 2021; Oreshkin et al., 2019). Early deep learning based approaches built upon a standard MLP models (Oreshkin et al., 2019) or recurrent networks such as LSTMs (Salinas et al., 2020). Recently, temporal convolution (Yue et al., 2021) and transformer (Li et al., 2019; Xu et al., 2021) networks have shown promising results, achieving promising on a wide range of real-world time series. *However, such methods assume a static world and information to forecast the future is fully provided in the look-back window. As a result, they lack the ability to remember events beyond the look-back window and adapt to the changing environments on the fly.* In contrast, our FSNet framework addresses these limitation by a novel adapter and memory components.

## B.2  CONTINUAL LEARNING

Human learning has inspired the design of several strategies to enable continual learning in neural networks. One successful framework is the *Complementary Learning Systems* theory (McClelland et al., 1995; Kumaran et al., 2016) which decomposes learning into two processes of learning fast (hippocampus) and slow (neocortex). While the hippocampus can quickly change and capture the current information, possibly with the help of experience replay, the neocortex does not change as fast and only accumulate general knowledge. The two learning systems interacts via a knowledge consolidation process, where recent experiences in the hippocampus are transferred to the neocortex to form a more general representation. In addition, the hippocampus also queries information from the neocortex to facilitate the learning of new and recurring events. The CLS theory serves as a motivation for several designs in continual learning such as experience replay (Chaudhry et al., 2019), dual learning architectures (Pham et al., 2021a; Arani et al., 2021). In this work, we extend the fast and slow learning framework of the CLS theory to the online time series forecasting problem.

## B.3  COMPARISON WITH EXISTING CONTINUAL LEARNING FOR TIME SERIES FORMULATIONS

This section provides a more comprehensive comparison between our formulation of online time series forecasting and existing studies in (He & Sick, 2021; Jaeger, 2017; Gupta et al., 2021; Kurle et al., 2019).

We first summarize the scope of our study. We mainly concern the online time series forecasting problem (Liu et al., 2016) and focus on addressing the challenge of fast adaptation to distribution shifts in this scenario. Particularly, when such distribution shifts happen, the model is required to take less training samples to achieve low errors, either by exploiting its representation capabilities or reusing the past knowledge. We focus on the class of deep feedforward neural network, particularly TCN, thanks to its powerful representation capabilities and ubiquitous in sequential data applications (Bai et al., 2018).

CLeaR (He & Sick, 2021) also attempted to model time series forecasting as a continual learning problem. However, CLeaR focuses on accumulating knowledge over a data stream without forgetting and does not concern about a fast adaptation under distribution shifts. Particularly, CLeaR's online training involves **periodically** calibrating the pre-trained model on new out-of-distribution samples using a continual learning strategy. Moreover, CLeaR only calibrates the model when a

buffer of novel samples are filled. As a result, when a distribution shifts, CLeaR could suffer from arbitrary high errors until it accumulates enough samples for calibrating. Therefore, CLeaR is not applicable to the online time series forecasting problem considered in our study.

GR-IG (Gupta et al., 2021) also formulates time series forecasting as a continual learning problem. However, they address the challenging of variable input dimensions through time, which could arise from the introduction of new sensors, or sensor failures. Therefore, by motivating from continual learning, GR-IG can facilitate the learning of new tasks (sensors) for better forecasting. However, GR-IG does not consider shifts in the observed distributions and focus on learning new distributions that appear over time. Consequently, GR-IG is also not a direct comparison to our method.

Lastly, we also note Conceptors (Jaeger, 2017) as a potential approach to address the time series forecasting problem. Conceptors are a class of neural memory that supports storing and retrieving patterns learned by a recurrent network. In this work, we choose to use the associative memory to maintain long-term patterns, which is more common for deep feed-forward architectures used in our work. We believe that with necessary adaptation, it is possible to integrate Conceptors as the memory mechanism in FSNet, which is beyond the scope of this work.

## C   FSNet Details

### C.1   Chunking Operation

In this section, we describe the chunking adapter's chunking operation to efficiently compute the adaptation coefficients. For convenient, we denote $\text{vec}(\cdot)$ as a vectorizing operation that flattens a tensor into a vector; we use $\text{split}(e, B)$ to denote splitting a vector $e$ into $B$ segments, each has size $\dim(e)/B$. An adapter maps its backbone's layer EMA gradient to an adaptation coefficient $u \in \mathbb{R}^d$ via the chunking process as:

$$
\begin{aligned}
\hat{g}_l &\leftarrow \text{vec}(\hat{g}_l) \\
[b_1, b_2, \ldots b_d] &\leftarrow \text{reshape}(\hat{g}_l; d) \\
[h_1, h_2, \ldots, h_d] &\leftarrow [W_\phi^{(1)} b_1, W_\phi^{(1)} b_2, \ldots, W_\phi^{(1)} b_d] \\
[u_1, u_2, \ldots, u_d] &\leftarrow [W_\phi^{(2)} h_1, W_\phi^{(2)} h_2, \ldots, W_\phi^{(2)} h_d].
\end{aligned}
$$

Where we denote $W_\phi^{(1)}$ and $W_\phi^{(2)}$ as the first and second weight matrix of the adapter. In summary, the chunking process can be summarized by the following steps: (1) flatten the gradient EMA into a vector; (2) split the gradient vector into $d$ chunks; (3) map each chunk to a hidden representation; and (4) map each hidden representation to a coordinate of the target adaptation parameter $u$.

### C.2   FSNet Pseudo Alorithm

Algorithm 1 provides the psedo-code for our FSNet.

---

**Algorithm 1** Fast and Slow learning Networks (FSNet)

---

**Require:** Two EMA coefficients $\gamma' < \gamma$, memory interaction threshold $\tau$
**Init:** backbone $\boldsymbol{\theta}$, adapter $\boldsymbol{\phi}$, associative memory $\mathcal{M}$, regressor $\boldsymbol{R}$, trigger = False

1   **for** $t \leftarrow 1$ **to** $T$ **do**
2      Receive the $t-$ look-back window $\boldsymbol{x}_t$
3      $\boldsymbol{h}_0 = \boldsymbol{x}_t$
4      **for** $j \leftarrow 1$ **to** $L$ **do**                              // Forward computation over $L$ layers
5         $[\boldsymbol{\alpha}_l, \boldsymbol{\beta}_l] = \boldsymbol{u}_l$, where $\boldsymbol{u}_l = \boldsymbol{\Omega}(\hat{\boldsymbol{g}}_l; \boldsymbol{\phi}_l)$                 // Initial adaptation parameter
6         **if** trigger == True **then**
7            $\tilde{\boldsymbol{u}}_l \leftarrow \mathrm{Read}(\hat{\boldsymbol{u}}_l, \mathcal{M}_l)$
8            $\mathcal{M}_l \leftarrow \mathrm{Write}(\mathcal{M}_l, \hat{\boldsymbol{u}}_l)$           // Memory read and write are defined in Section 3.2.2
9            $\boldsymbol{u}_l \leftarrow \tau \boldsymbol{u}_l + (1 - \tau)\tilde{\boldsymbol{u}}_l$       // Weighted sum the current and past adaptation parameters
10        $\tilde{\boldsymbol{\theta}}_l = \mathrm{tile}(\boldsymbol{\alpha}_l) \odot \boldsymbol{\theta}_l$                                // Weight adaptation
11        $\tilde{\boldsymbol{h}}_l = \mathrm{tile}(\boldsymbol{\beta}_l) \odot \boldsymbol{h}_l$, where $\boldsymbol{h}_l = \tilde{\boldsymbol{\theta}}_l \circledast \tilde{h}_{l-1}$.             // Feature adaptation
12      Forecast $\hat{\boldsymbol{y}}_t = \boldsymbol{R}h_T$
13      Receive the ground-truth $\boldsymbol{y}$
14      Calculate the forecast loss and backpropagate
15      Update the regressor $\boldsymbol{R}$ via SGD
16      **for** $j \leftarrow 1$ **to** $L$ **do**                              // Backward to update the model and EMA
17         Update the EMA of $\hat{\boldsymbol{g}}_l, \hat{\boldsymbol{g}}_l', \boldsymbol{u}_l$
18         Update $\boldsymbol{\phi}_l, \boldsymbol{\theta}_l$ via SGD
19         **if** $\cos(\hat{\boldsymbol{g}}_l, \hat{\boldsymbol{g}}_l) < -\tau$ **then**
20            trigger $\leftarrow$ True

---

## D   EXPERIMENT DETAILS

### D.1   SYNTHETIC DATA

We use the following first-order auto-regressive process model $AR_\varphi(1)$ defined as

$$X_t = \varphi X_{t-1} + \epsilon_t, \tag{7}$$

where $\epsilon_t$ are random noises and $X_{t-1}$ are randomly generated. The S-Abrupt data is described by the following equation:

$$X_t = \begin{cases} AR_{0.1} & \text{if } 1 < t \le 1000 \\ AR_{0.4} & \text{if } 1000 < t \le 1999 \\ AR_{0.6} & \text{if } 2000 < t \le 2999 \\ AR_{0.1} & \text{if } 3000 < t \le 3999 \\ AR_{0.4} & \text{if } 4000 < t \le 4999 \\ AR_{0.6} & \text{if } 5000 < t \le 5999. \end{cases} \tag{8}$$

The S-Gradual data is described as

$$X_t = \begin{cases} AR_{0.1} & \text{if } 1 < t \le 800 \\ 0.5 \times (AR_{0.1} + AR_{0.4}) & \text{if } 800 < t \le 1000 \\ AR_{0.4} & \text{if } 1000 < t \le 1600 \\ 0.5 \times (AR_{0.4} + AR_{0.6}) & \text{if } 1600 < t \le 1800 \\ AR_{0.6} & \text{if } 1800 < t < 2400 \\ 0.5 \times (AR_{0.6} + AR_{0.1}) & \text{if } 2400 < t \le 2600 \\ AR_{0.1} & \text{if } 2600 < t \le 3200 \\ 0.5 \times (AR_{0.1} + AR_{0.4}) & \text{if } 3200 < t \le 3400 \\ AR_{0.4} & \text{if } 3400 < t \le 4000 \\ 0.5 \times (AR_{0.4} + AR_{0.6}) & \text{if } 4000 < t \le 4200 \\ AR_{0.6} & \text{if } 4200 < t \le 5000 \end{cases} \tag{9}$$

### D.1.1 BASELINE DETAILS

**Summary**  We provide a brief summary of the baselines used in your experiments

- **Informer** (Zhou et al., 2021): a transformer-based model for time-series forecasting.
- **OnlineTCN** uses a standard TCN backbone (Woo et al., 2022) with 10 hidden layers, each of which has two stacks of residual convolution filters.
- **ER** (Chaudhry et al., 2019) augments the OnlineTCN baseline with an episodic memory to store previous samples, which are then interleaved when learning the newer ones.
- **MIR** (Aljundi et al., 2019a) replaces the random sampling strategy in ER with its MIR sampling by selecting samples in the memory that cause the highest forgetting and perform ER on these samples.
- **DER++** (Buzzega et al., 2020) augments the standard ER (Chaudhry et al., 2019) with a $\ell_2$ knowledge distillation loss on the previous logits.
- **TFCL** (Aljundi et al., 2019b) is a method for online, task-free continual learing. TFCL starts with as a ER procedure and also includes a MAS-styled (Aljundi et al., 2018) regularization that is adapted for the task-free setting.

All ER-based strategies use a reservoir sampling buffer. We also tried with a Ring buffer and did not observe any significant differences.

**Loss function**  All methods in our experiments optimize the $\ell_2$ loss function defined as follows. Let $\boldsymbol{x}$ and $\boldsymbol{y} \in \mathbb{R}^H$ be the look-back and ground-truth forecast windows, and $\hat{\boldsymbol{y}}$ be the model's prediction of the true forecast windows. The $\ell_2$ loss is defined as:

$$\ell(\hat{\boldsymbol{y}}_t, \boldsymbol{y}_t) = \ell(f_{\boldsymbol{\omega}}(\boldsymbol{x}_t), \boldsymbol{y}_t) \coloneqq \frac{1}{H} \sum_{j=1}^{H} ||\hat{\boldsymbol{y}}_i - \boldsymbol{y}_i||^2 \tag{10}$$

**Experience Replay baselines**  We provide the training details of the ER and DER++ baselines in this section. These baselines deploy an reservoir sampling buffer of 500 samples to store the observed samples (each sample is a pair of look-back and forecast window).

Let $\mathcal{M}$ be the episodic memory storing previous samples, $\mathcal{B}_t$ be a mini-batch of samples sampled from $\mathcal{M}$. ER minimizes the following loss function:

$$\mathcal{L}_t^{\text{ER}} = \ell(f_{\boldsymbol{\omega}}(\boldsymbol{x}_t), \boldsymbol{y}_t) + \lambda_{\text{ER}} \sum_{(\boldsymbol{x}, \boldsymbol{y}) \in \mathcal{B}_t} \ell(f_{\boldsymbol{\omega}}(\boldsymbol{x}), \boldsymbol{y}), \tag{11}$$

where $\ell(\cdot, \cdot)$ denotes the MSE loss and $\lambda_{\text{ER}}$ is the trade-off parameter of current and past examples. DER++ further improves ER by adding a distillation loss (Hinton et al., 2015). For this purpose, DER++ also stores the model's forecast into the memory and minimizes the following loss:

$$\mathcal{L}_t^{\text{DER++}} = \ell(f_{\boldsymbol{\omega}}(\boldsymbol{x}_t), \boldsymbol{y}_t) + \lambda_{\text{ER}} \sum_{(\boldsymbol{x}, \boldsymbol{y}) \in \mathcal{B}_t} \ell(f_{\boldsymbol{\omega}}(\boldsymbol{x}), \boldsymbol{y}) + \lambda_{\text{DER++}} \sum_{(\boldsymbol{x}, \hat{\boldsymbol{y}}) \in \mathcal{B}_t} \ell(f_{\boldsymbol{\omega}}(\boldsymbol{x}), \hat{\boldsymbol{y}}). \tag{12}$$

### D.2 HYPER-PARAMETERS SETTINGS

We cross-validate the hyper-parameters on the ETTh2 dataset and use it for the remaining ones. Particularly, we use the following configuration:

- Adapter's EMA coefficient $\gamma = 0.9$,
- Gradient EMA for triggering the memory interaction $\gamma' = 0.3$
- Memory triggering threshold $\tau = 0.75$

We found that this hyper-parameter configuration matches the motivation in the development of FSNet. In particular, the adapter's EMA coefficient $\gamma = 0.9$ can capture medium-range information to facilitate the current learning. Second, the gradient EMA for triggering the memory interaction

Table 3: Standard deviations of the metrics in Table 1. "†" indicates a transformer backbone, "-" indicates the model did not converge. S-A: S-Abrupt, S-G: S-Gradual.

| Method | | FSNet | | DER++ | | MIR | | ER | | TFCL | | OnlineTCN | | Informer | |
|---|---|---|---|---|---|---|---|---|---|---|---|---|---|---|---|
| | H | MSE | MAE | MSE | MAE | MSE | MAE | MSE | MAE | MSE | MAE | MSE | MAE | MSE | MAE |
| ETTh2 | 1 | 0.018 | 0.009 | 0.022 | 0.015 | 0.019 | 0.018 | 0.018 | 0.017 | 0.030 | 0.003 | 0.011 | 0.007 | 1.370 | 0.043 |
| | 24 | 0.014 | 0.005 | 0.024 | 0.004 | 0.017 | 0.005 | 0.007 | 0.006 | 0.005 | 0.003 | 0.017 | 0.002 | 2.254 | 0.102 |
| | 48 | 0.128 | 0.012 | 0.143 | 0.015 | 0.130 | 0.012 | 0.141 | 0.013 | 0.279 | 0.024 | 0.147 | 0.016 | 2.088 | 0.091 |
| ETTm1 | 1 | 0.003 | 0.004 | 0.003 | 0.007 | 0.005 | 0.009 | 0.005 | 0.009 | 0.004 | 0.008 | 0.003 | 0.002 | 0.088 | 0.060 |
| | 24 | 0.002 | 0.002 | 0.002 | 0.002 | 0.005 | 0.004 | 0.003 | 0.002 | 0.006 | 0.005 | 0.002 | 0.002 | 0.035 | 0.023 |
| | 48 | 0.003 | 0.002 | 0.003 | 0.002 | 0.006 | 0.005 | 0.004 | 0.004 | 0.010 | 0.008 | 0.002 | 0.003 | 0.020 | 0.014 |
| ECL | 1 | 0.021 | 0.001 | 0.027 | 0.002 | 0.037 | 0.013 | 0.034 | 0.011 | 0.047 | 0.011 | 0.019 | 0.002 | - | - |
| | 24 | 0.096 | 0.011 | 0.072 | 0.013 | 0.261 | 0.013 | 0.236 | 0.017 | 0.338 | 0.019 | 0.077 | 0.009 | - | - |
| | 48 | 0.105 | 0.011 | 0.146 | 0.014 | 0.143 | 0.012 | 0.320 | 0.014 | 0.253 | 0.008 | 0.122 | 0.011 | - | - |
| Traffic | 1 | 0.001 | 0.001 | 0.001 | 0.001 | 0.001 | 0.001 | 0.001 | 0.001 | 0.004 | 0.003 | 0.001 | 0.001 | 0.009 | 0.008 |
| | 24 | 0.002 | 0.002 | 0.002 | 0.002 | 0.002 | 0.002 | 0.002 | 0.002 | 0.004 | 0.002 | 0.002 | 0.001 | 0.015 | 0.008 |
| WTH | 1 | 0.001 | 0.001 | 0.001 | 0.002 | 0.002 | 0.002 | 0.001 | 0.002 | 0.002 | 0.002 | 0.001 | 0.001 | 0.005 | 0.005 |
| | 24 | 0.001 | 0.001 | 0.001 | 0.001 | 0.001 | 0.001 | 0.001 | 0.001 | 0.001 | 0.001 | 0.001 | 0.001 | 0.003 | 0.003 |
| | 48 | 0.001 | 0.001 | 0.011 | 0.007 | 0.009 | 0.005 | 0.009 | 0.005 | 0.004 | 0.006 | 0.001 | 0.001 | 0.009 | 0.008 |
| S-A | 1 | 0.112 | 0.037 | 0.171 | 0.041 | 0.176 | 0.040 | 0.159 | 0.033 | 0.283 | 0.061 | 0.009 | 0.002 | 0.149 | 0.059 |
| | 24 | 0.199 | 0.027 | 0.202 | 0.034 | 0.192 | 0.032 | 0.022 | 0.003 | 0.011 | 0.002 | 0.174 | 0.017 | 0.322 | 0.066 |
| S-G | 1 | 0.166 | 0.039 | 0.169 | 0.041 | 0.177 | 0.040 | 0.171 | 0.039 | 0.360 | 0.083 | 0.164 | 0.039 | 0.277 | 0.099 |
| | 24 | 0.187 | 0.033 | 0.200 | 0.033 | 0.199 | 0.035 | 0.190 | 0.032 | 0.010 | 0.002 | 0.188 | 0.033 | 0.771 | 0.099 |

$\gamma' = 0.3$ results in the gradients accumulated in only a few recent samples. Lastly, a relatively high memory triggering threshold $\tau = 0.75$ indicates our memory-triggering condition can detect substantial representation change to store in the memory. The hyper-parameter cross-validation is performed via grid search and the grid is provided below.

- Experience replay batch size (for ER and DER++): $[2, 4, 8]$

- Experience replay coefficient (for ER) $\lambda_{ER}$: $[0.1, 0.2, 0.5, 0.7, 1]$

- DER++ coefficient (for DER++) $\lambda_{DER++}$: $[0.1, 0.2, 0.5, 0.7, 1]$

- EMA coefficient for FSNet $\gamma$ and $\gamma'$: $[0.1, 0.2, 0.3, 0.4, 0.5, 0.6, 0.7, 0.8, 0.9]$

- Memory triggering threshold $\tau$: $[0.6, 0.65, 0.7, 0.75, 0.8, 0.85, 0.9]$

- Number of filters per layer: 64

- Episodic memory size: 5000 (for ER, MIR, and DER++), 50 (for TFCL)

The remaining configurations such as data pre-processing and optimizer setting follow exactly as Zhou et al. (2021).

# E ADDITIONAL RESULTS

## E.1 STANDARD DEVIATIONS

We report the standard deviation values of the comparison experiment in Table 1, which were averaged over five runs. Overall, we observe that the standard deviation values are quite small for all experiments.

## E.2 COMPLEXITY COMPARISON

In this Section, we analyze the memory and time complexity of FSNet.

Table 4: Summary of the model complexity on the ETTh2 data set with forecasta window $H = 24$. We report the number of floating points incurred by the backbone and different types of memory. GI = Gradient Importance (TFCL), G-EMA = Gradient Exponential Moving Average (FSNet), AM = Associative Memory (FSNet), EM = Episodic Memory (ER).

| Method | Model | | Memory | | | | Total |
| --- | --- | --- | --- | --- | --- | --- | --- |
| | Backbone | Adapter | GI | G-EMA | AM | EM | |
| FSNet | 1,041,288 | 733,334 | N/A | 614,400 | 1,130,496 | N/A | 3,519,518 |
| ER | 1,041,288 | N/A | N/A | N/A | N/A | 2,822,400 | 3,863,688 |
| OnlineTCN | 3,667,208 | N/A | N/A | N/A | N/A | N/A | 3,667,208 |
| TFCL | 1,041,288 | N/A | 2,082,576 | N/A | N/A | 806,400 | 3,930,264 |

Table 5: Summary of the model and total memory complexity of different methods. $N$ denotes the number parameters of the convolutional layers, $H$ and $E$ denotes the look-back and forecast windows length

| Method | OnlineTCN | ER | MIR | DER++ | FSNet |
| --- | --- | --- | --- | --- | --- |
| Model Complexity | | $\mathcal{O}(N + H)$ | | | |
| Memory Complexity | N/A | $\mathcal{O}(E + H)$ | | | $\mathcal{O}(N)$ |
| Total Complexity | $\mathcal{O}(N + H)$ | $\mathcal{O}(N + E + H)$ | | | $\mathcal{O}(N + H)$ |

**Asymptotic analysis**   We consider the TCN forecaster used throughout this work and analyze the *model, total memory, and time* complexities of the methods considered in our work. We let $N$ denotes the number of parameters of the the convolutional layers, $E$ denotes the length of the look-back window, and $H$ denotes the length of the forecast window.

**Model and Total complexity**   We analyze the model and the total memory complexity, which arises from the model and additional memory units.

First, the standard TCN forecaster incur a $\mathcal{O}(N + H)$ memory complexity arising from $N$ parameters of the convolutional layers, and an order of $H$ parameters from the linear regressor.

Second, we consider the replayed-based strategies, which also incur the same $\mathcal{O}(N + H)$ model complexity as the OnlineTCN. For the total memory, they use an episodic memory to store the previous samples, which costs $\mathcal{O}(E + H)$ for both methods. Additionally, TFCL stores the importance of previous parameters while MIR makes a copy of the model for its virtual update, both of which cost $\mathcal{O}(N + H)$. Therefore, the total memory complexity of the replay strategies (ER, DER++, MIR, and TFCL) is $\mathcal{O}(N + E + H)$.

Third, in FSNet, both the per-layer adapters and the associative memory cost similar number of parameters as the convolutional layers because they are matrices with number of channels as one dimension. Therefore, asymptotically, FSNet also incurs a model and total complexity of $\mathcal{O}(N + H)$ where the constant term is small.

Table 5 summarizes the asymptotic memory complexity discussed so far. Table 4 shows the number of parameters used of different strategies on the ETTh2 dataset with the forecast window of $H = 24$. We consider the total parameters (model and memory) of FSNet as the total budget and adjust other baselines to meet the budget. As we analyzed, for FSNet, its components, including the adapter, associative memory, and gradient EMA, require an order of parameter as the convolutional layers in the backbone network. For the OnlineTCN strategy, we increases the number of convolutional filters so that it has roughly the same total parameters as FSNet. For ER and TFCL, we change the number of samples stored in the episodic memory.

**Time Complexity**   We report the throughput (samples/second) of different methods in Table 6. We can see that ER and DER++ have high throughput (low running time) compared to others thanks to their simplicity. As FSNet introduces additional mechanisms to allow the network to take less samples to adapt to the distribution shifts, its throughput is lower than ER and DER++. Neverthe-

Table 6: Throughput (sample/second) of different methods in our experiments with forecast window of $H = 1$.

| Running Time | ETTh2 | ETTm1 | WTH | ECL | Traffic | S-A |
|---|---|---|---|---|---|---|
| ER | 46 | 46 | 43 | 42 | 39 | 46 |
| DER++ | 45 | 45 | 43 | 42 | 38 | 46 |
| TFCL | 29 | 28 | 27 | 27 | 26 | 27 |
| MIR | 22 | 22 | 21 | 21 | 30 | 23 |
| FSNet | 28 | 28 | 28 | 27 | 27 | 29 |

Table 7: Results of different FSNet's hyper-parameter configurations on the ETTh2 ($H = 48$) and S-A ($H = 24$) benchmarks.

| Configuration | | | ETTh2 | | S-A | |
|---|---|---|---|---|---|---|
| $\gamma$ | $\gamma'$ | $\tau$ | MSE | MAE | MSE | MAE |
| 0.9 | 0.3 | 0.75 | 0.846 | 0.515 | 1.760 | 1.038 |
| 0.9 | 0.4 | 0.8 | 0.860 | 0.521 | 1.816 | 1.086 |
| 0.99 | 0.4 | 0.7 | 0.847 | 0.512 | 1.791 | 1.049 |
| 0.99 | 0.3 | 0.8 | 0.845 | 0.514 | 1.777 | 1.042 |

less, FSNet is more efficient than and MIR comparable to TFCL, which are two common continual learning strategies.

### E.3 ROBUSTNESS OF HYPER-PARAMETER SETTINGS

This experiment explores the robustness of FSNet to different hyper-parameter setting. Particularly, we focus on the configuration of *three* hyper-parameters: (i) the gradient EMA $\gamma$; (ii) the short-term gradient EMA $\gamma'$; and (iii) the associative memory activation threshold $\tau$. In general, we provide two guidelines to reduce the search space of these hyper-parameters: (i) setting $\gamma$ to a high value (e.g. 0.9) and $\gamma'$ to a small value (e.g. 0.3 or 0.4); (ii) set $\tau$ to be relatively high (e.g. 0.75). We report the results of several hyper-parameter configurations in Table 7. We observe that there are not significant differences among these configurations . It is also worth noting that we use the same configuration for all experiments conducted in this work. Therefore, we can conclude that FSNet is robust to these configurations.

### E.4 FSNET AND EXPERIENCE REPLAY

This experiment explore the complementarity between FSNet and experience replay (ER). We hypothesize that ER is a valuable component when learning on data streams because it introduces the benefits of mini-batch training to online learning.

We implement a variant of FSNet with an episodic memory for experience replay and report its performance in Table 8. We can see that FSNet+ER outperforms FSNet in all cases, indicating the benefits of ER, even to FSNet. However, it is important that using ER will introduce additional memory complexity and that scales with the look-back window. Lastly, in many real-world applications, storing previous data samples might be prohibited due to privacy concerns.

### E.5 VISUALIZATIONS

#### E.5.1 VISUALIZATION OF THE SYNTHETIC DATASETS

We plot the raw data (before normalization) of the S-Abrupt and S-Gradual datasets in Figure 4.

Table 8: Performance of FSNet with and without experience replay.

| Data | H | FSNet | | FSNet+ER | |
|---|---|---|---|---|---|
| | | MSE | MAE | MSE | MAE |
| ETTh2 | 1 | 0.466 | 0.368 | **0.434** | **0.361** |
| | 24 | 0.687 | 0.467 | **0.650** | **0.462** |
| | 48 | 0.846 | 0.515 | **0.842** | **0.511** |
| Traffic | 1 | 0.321 | 0.26 | **0.243** | **0.248** |
| | 24 | 0.421 | 0.312 | **0.350** | **0.275** |

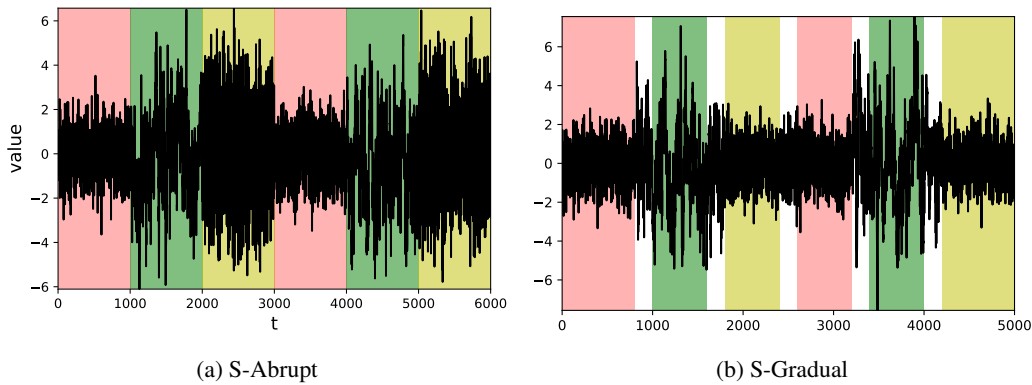

(a) S-Abrupt        (b) S-Gradual

Figure 4: Visualization of the raw S-Abrupt and S-Gradual datasets before normalization. Colored regions indicate the data generating distribution where we use the same color for the same distribution. In S-Guadual, white color region indicates the gradual transition from one distribution to another. Best viewed in color.

### E.5.2   ACTIVATION PATTERN OF FSNET

This experiment explores the associative memory activation patterns of FSNet. For this, we consider the S-Abrupt dataset with $H = 1$ and plot the activation patterns in Figure 5. Note that due to the large number of memory slots, we only plot the memory slot with the highest attention score at each step. We remind that in S-Abrupt, the first 3,000 samples belong to three different data distribution and these distribution sequentially reappear in the last 3,000 samples, which are color-coded in Figure 5. First, we observe that not all layers are equally important for the tasks. Particularly, FSNet mostly uses the fourth and sixth layers, and rarely uses the deeper ones.

Second, we note that FSNet memory activations exhibit high specialization as we go to deeper layers. Particularly, only a single memory slot is activated in the fourth layer (circle marker) throughout training because shallow layers are responsible for general representations, possibly because it learns generic representations for all patterns. On the other hand, deeper layers are activated according to different distributions: seventh layer memory (triangle marker) is activated by the distribution in pink while the ninth layer memory (square and star markers) is activated by the remaining distributions. These observations are consistent with the representation learning patterns in deep networks where shallow layers learn generic representation while deeper layers learn representations that are more specialized to different patterns (Olah et al., 2017).

## F   DISCUSSION AND FUTURE WORK

We discuss two scenarios where FSNet may not work well. First, we suspect that FSNet may struggle when concept drifts do not happen uniformly on all dimensions. This problem arises from the irregularly sampled time series, where each dimension is sampled at a different rate. In this scenario, a concept drift in one dimension may trigger FSNet's memory interaction and affect the

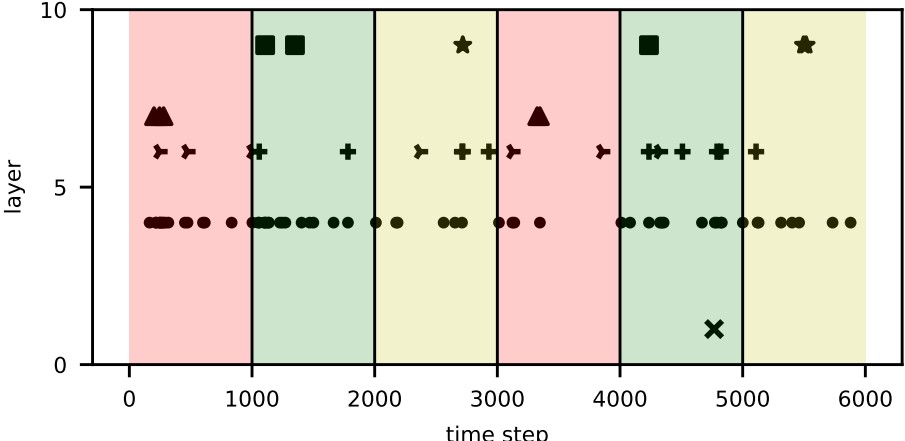

Figure 5: Activation frequency of the memory slot with the highest attention score for each layer in FSNet on the S-Abrupt dataset. Same marker indicates the same memory slot. Each color region indicates a data generating distribution. Best viewed in color.

learning of the remaining ones. Moreover, if a dimension is sampled too sparsely, it might be helpful to leverage the relationship along both the time and spatial dimension for a better result.

Second, applications such as finance, which involve many complex repeating patterns, can be challenging for FSNet. In such cases, the number of repeating patterns may exceed the memory capacity of FSNet, causing catastrophic forgetting. In addition, forecasting complex time series requires the network to learn a good representation, which may not be achieved by increasing the model complexity alone. In such cases, incorporating a representation learning component might be helpful.

We now discuss several aspects for further studies. We follow Informer to apply the z-normalization per feature, which is a common strategy. This strategy works well in the batch setting because its statistics were estimated using 80% of training data. However, after a concept drift in online learning, it is unreliable to use previous statistics (estimated over 25% samples) to normalize samples from a new distribution. In such cases, it could be helpful to adaptively normalize samples from new distributions (using the new distribution's statistics). This could be achieved via an online update of the normalization statistics or using a sliding window technique. In addition, while FSNet presents a general framework to forecast time series online, adopting it to a particular application requires incorporating specific domain knowledge to ensure satisfactory performances. In summary, we firmly believe that FSNet is an encouraging first step towards a general solutions for an important, yet challenging problem of online time series forecasting.

