# OpenReview forum: "Learning Fast and Slow for Online Time Series Forecasting"
_ICLR.cc/2023/Conference — ICLR 2023 poster_

### Official Review · Reviewer_MVEA · 2022-10-25

**Confidence:** 3
**Correctness:** 3
**Technical Novelty And Significance:** 3
**Empirical Novelty And Significance:** 3
**Recommendation:** 6

**Clarity, Quality, Novelty And Reproducibility:**

The paper is well written and easy to follow. Technical details are clear and solid. The presented adaptation and memory components, while intuitive, are novel. Public code will increase reproducibility of the work.


**Strength And Weaknesses:**

Strengths:

- The proposed mechanisms of fast adaptation and recalling recurring patterns are intuitive and interesting.

- Discarding the need of task boundaries is interesting and useful, although there are some technical questions regarding how the model may handle lookback window/forecasting horizons that contain "unknown" tasks transitions.

- The experiments considered a variety of benchmark data as well as synthetic data with controlled transitions of tasks. The comparisons considered a good number of representative baselines demonstrating margins of improvement.



Weakness:

- A key question is regarding the size of the memory needed in the different datasets/experiments. How is the necessary size determined, and does the memory expand? In synthetic settings where ground truth of the tasks are known, it also be good to know what were learned in M, whether it corresponds to the four true processes, and whether the correct adaptation coefficients were retrieved when the old pattern reappeared.


- How much the key hyperprameters, such as tao that triggers the member association in Eq (3), depends on the underlying datasets and what are their effects


- From Fig 2, it was not clear if the presented method indeed achieved "faster" adaptation. It seemed that the error trend is more or less the same across methods as learning continues. The presented method was overall better, but when there was an increase of error, it was not clear that the presented method had a "faster" drop of errors -- the trend is similar (how fast the error drops); it's just that the presented method had overall lower errors. Please clarify if my understanding was not correct.

- It was not clear how the model may work when the look-back window has a mix of tasks (in between transitions). The S-G setup somewhat reflects this, but even in S-A, such scenario exists (when the look-back window or forecasting horizon contains the task transition boundary), correct? How would the presented method accommodate this? By learning and memorizing many different adaptation coefficients reflecting the transition in between tasks?

- In both Fig 2 and Fig 3, for the data where tasks transition are known (eg. the two synthetic sets), it'd be good to label the boundary in the figure (even though one can track it from the texts, i.e., every 1000 steps, it'd be good to make it more intuitive on the figures).

-The separation of mean and standard deviation in the main text and appendix makes it quite hard to follow and assess whether the obtained margins of improvements were statistically significant. This is an important aspect of model performance, and it'd be good to try to move the std in the main text as well.





**Summary Of The Paper:**

This paper approaches online time series from the angle of task-free continual learning. This was achieved with a fast adaptation mechanism utilizing a moving average of the model gradient, and an associative memory module to recall the adaptation coefficients of reoccurring events. Experiments were performed on both benchmark data and synthesized datasets designed with explicit task changes, in comparison to a good number of baseline models.


**Summary Of The Review:**

This paper presents a continual learning approach to online time-series modeling and forecasting, with two interesting components that are intuitive and well described. There were some lingering technical questions to be clarified, but the overall paper seems to be interesting with thorough experimental evaluations

---

> ### Author Response · Authors · 2022-11-18
> **Response to Reviewer MVEA (part 1)**
>
> **Concern #1: “size of the memory needed in the experiments. How is the necessary size determined, and does the memory expand”**
>
> Thank you for the interesting question. All experiments in Table 1 use a fixed memory size of M=32.
> We also conducted an ablation study that varies the memory size in Table 2, ranging from not using the memory at all, to a large memory of M=128 per layer. Overall, we observe that the associative memory offers improvements when the dataset contains sudden distribution shifts (ETTh2 or S-A). On the other hand, when the distribution shifts more gradually (Traffic or S-G), the associative memory’s impact is more marginal because the adapter alone can handle such shifts.
>
> In our work, we choose the memory size of M=32 such that the total complexity (model + memory) is the same for all methods (please refer to Appendix E.2 for a detailed analysis of memory and time complexity). In general, increasing the memory size will yield better performance (as shown in Table 2). Choosing the optimal memory sizes will depend on each dataset and require carefully analyzing each dimension in the time series data, which is extremely costly and time-consuming.
>
> **Concern #2 “, it also be good to know what were learned in M, whether it corresponds to the four true processes, and whether the correct adaptation coefficients were retrieved when the old pattern reappeared.”**
>
> We plot the activation pattern of the memory slots with the highest attention score in Figure 5, Appendix E.5.2. Overall, we observe that FSNet memory activations exhibit high specialization as we go to deeper layers. Particularly, only a single memory slot is activated in the fourth layer (circle marker) throughout training because shallow layers are responsible for learning general representations from all patterns. On the other hand, deeper layers are activated according to different distributions: seventh layer’s memory (triangle marker) is activated by the distribution in pink while the ninth layer’s memory (square and star markers) is activated by the remaining distributions. These observations are consistent with the representation learning patterns in deep networks where shallow layers learn generic representation while deeper layers learn representations that are more specialized to different patterns.
>
> **Concern #3: “How much the key parameters depend on each dataset and what are their effects?”**
>
> FSNet introduces three additional hyper-parameters. First, $\tau$: the memory activation threshold, which determines the memory activation frequency. High values of $\tau$ constrain the memory to only activate to significant changes in the representation while lower values allow for a more frequent activation, which increases the running time and be more susceptible to noises.
>
> Second,  $\gamma$: the standard gradient exponential moving average (EMA), used to capture the temporal smoothness in time-series, which is used as input to the adapter.
> Lastly,  $\gamma’$: the short-term gradient EMA, which is used together with $\gamma$ to detect a representation change.
>
> In Appendix E.3, we provided a guideline for choosing these hyper-parameter based on their functionalities: (i) setting $\gamma$ to a high value (e.g. 0.8) and $\gamma’$ to a relatively low value (e.g. 0.3); and (ii) setting $\tau$ to a relative high value (e.g. 0.75).
>
> We further showed that FSNet is robust to the configuration of these hyper-parameters. Table 7, Appendix E.3, reports FSNet’s performance under different configurations. The results show that there are no significant differences amongst different configurations, as long as they follow our guideline. Lastly, it is also worth noting that FSNet uses the same hyper-parameter configuration on all datasets reported in Table 1. These results support that FSNet is robust to the hyper-parameter configurations.
>
> **Concern #4: “it was not clear if the presented method indeed achieved ‘faster’ adaptation”; “ it was not clear that the presented method had a "faster" drop of errors”**
>
> Our FSNet achieved faster learning by having lower error peaks whenever the distribution changed, which can be observed in Figure 2.
> Whenever a distribution shift happens, the models need to learn new patterns, causing the errors to increase. Without a fast learning mechanism, the baselines’ errors can grow arbitrarily high before they can adapt. In contrast, our FSNet could start learning soon after the distribution shifts, thus having lower error curves. Another argument for FSNet’s fast learning is that it needs fewer samples to achieve a certain error threshold, which can also be seen in Figure 2.
> This result is further supported by Figure 3, which visualizes the model’s forecast after a distribution shift in the S-A dataset. We can see FSNet can provide meaningful forecasts compared to other baselines.

---

> > ### Author Response · Authors · 2022-11-18
> > **Response to Reviewer MVEA (part 2)**
> >
> > **Concern #5: “It was not clear how the model may work when the look-back window has a mix of tasks”**
> >
> > Thank you for the interesting observation. FSNet’s behavior between task transitions depends on the type of distribution shifts (abrupt vs gradual).
> > For abrupt changes (S-A or ETTh2), the transition phase is discrete, making the representation changes substantially and triggers the associative memory to activate and recall the previous adaptation coefficients to support the adapter. Based on our observation from Figure 5, Appendix E.5.2, the memory activation patterns also vary across layers where shallow layers' memories seem to be more sensitive and are activated soon after a distribution shift, which may include the transition phase. On the other hand, deeper layers' memories are usually activated when the distribution is fully transitioned.
> >
> > For gradual changes (S-G), the representation may not change significantly enough to trigger the memory. Although the errors would increase initially, the adapter alone can adjust the base network to learn effectively.
> > This phenomenon is observed in Table 2. We can see that on the S-G dataset where the representation changes gradually, FSNet achieved similar performances when setting the memory to M=128, M=32, or M=0 (No Memory).  On the other hand, datasets with abrupt changes such as ETTh2 or S-A, using the associative memory provided more significant improvements.
> >
> > Therefore, we can conclude that the associative memory is helpful to deal with abrupt changes, while the adapter alone can successfully handle gradual changes.
> >
> > **Concern #6: In both Fig 2 and Fig 3, for the data where tasks transition are known (eg. the two synthetic sets), it'd be good to label the boundary in the figure (even though one can track it from the texts, i.e., every 1000 steps, it'd be good to make it more intuitive on the figures).**
> >
> > Thank you for the suggestion.
> > We have revised Figure 2(f) to include the indicators of when the distribution shifted.
> > Regarding Figure 3, we considered the S-A dataset and plot the models’ forecasts after 200 time steps after a repeating pattern occurs. As such, each row of Figure 3 corresponds to a period after distribution shifts. We have revised Figure 3 caption to discuss this more clearly.
> > We hope these changes will make the figure more intuitive and self-contained.
> >
> > **Concern #7: The separation of mean and standard deviation in the main text and appendix makes it quite hard to follow**
> >
> > We understand that splitting the mean and standard deviation would be hard to track if the improvements are significant. However, given the large scale experiments that we conducted, putting both numbers in the main paper would cause visual cluttering and it is not enough space to do so.
> > Fortunately, we observe that the differences between methods in Table 1 are significant. For example, in most cases, FSNet outperforms the second best baselines (DER++) by an order of 0.01 or more, while the standard deviations are in the order of 0.001. We have revised Table 1’s caption to mention the significance of the results.

---

> > > ### Comment · Reviewer_MVEA · 2022-11-29
> > > **Thanks for the response**
> > >
> > > I want to thank the author for the detailed response. The activation pattern of the memory slots, especially that of the deeper layers, is especially interesting. I wanted to confirm with the author that -- in response to question #5: in S-G settings, the look-back window may contain transition of samples in between tasks?

---

> > > > ### Author Response · Authors · 2022-11-29
> > > > **Response to follow-up questions from Reviewer MVEA**
> > > >
> > > > Dear Reviewer MVEA,
> > > >
> > > > Thank you for the response, please find our answer to your question below.
> > > >
> > > > **Q: the look-back window may contain transition of samples in between tasks**
> > > >
> > > > **A:** Yes, in the S-G dataset, the look-back windows contain transition of samples in between tasks.
> > > >
> > > > More specifically, after the task switches, the look-back windows (of length $e$) will have the form $(x_{i-e+1}, \ldots, x_k, \ldots ,x_i),  \text{with } k \leq i,$ where the two segments $(x_{i-e+1}, \ldots, x_k)$ and $(x_{k+1}, \ldots, x_i)$ are sampled from different distributions.

---

> ### Author Response · Authors · 2022-11-25
> **Further discussions**
>
> Dear Reviewer MVEA,
>
> Thank you again for the valuable comments and feedback. We hope that you have had time go through our response and the revised manuscript. We would like to check with you if our responses have been satisfactory. We look forward to further discuss with you if you have any additional comments.
>
> Best regards,
> Authors

---

> ### Author Response · Authors · 2022-12-05
> **Gentle Reminder**
>
> Dear Reviewer MVEA,
>
> Thanks for your time and valuable feedback! As the deadline for discussion is approaching, we look forward to seeing whether our responses properly address your concerns. In the original review, you raised several technical questions regarding the memory mechanism, hyper-parameter robustness, and FSNet's ability to fast learning, which we addressed in our initial response. We further confirmed your follow-up question that the look-back windows in our datasets contain samples from different distributions during transitions. We hope our responses have been satisfactory and clarified the novel aspects of FSNet, including its ability to learn fast and effectively, not requiring task boundaries, and its robustness to hyper-parameters.
>
> We hope you will consider this work a promising step toward addressing online time series forecasting and a practical application of continual learning.
>
> With best regards,
> Authors

---

### Official Review · Reviewer_r44V · 2022-10-26

**Confidence:** 4
**Correctness:** 3
**Technical Novelty And Significance:** 2
**Empirical Novelty And Significance:** 2
**Recommendation:** 6

**Clarity, Quality, Novelty And Reproducibility:**

This study is clearly explained and the work is novel in the investigated application.

**Strength And Weaknesses:**

The authors studied an interesting topic which is indeed critical in many practical cases.

The adaptation process that you explained in page 4 is somewhat very abstract and confusing (considering as your main contribution) and not even mentioned what for instance \phi_l or h_l refer to (one could find them by continue reading the paper, but could have been mentioned in their place).
As in the online case you want to have fast inference, I think it is necessary to include the exact inference time, for each method. Also a detailed analysis of the overhead memory for each method can be informative.

In my opinion, the novelty of this study is very limited, as the main aspect of this work has been already studied in the past  (Pham et al., 2021a; (Pham et al., 2021b, Arani et al., 2021). Basically, a continual learning scenario has been translated to an online learning scenario, where as in most cases actually there are a lot in common in these two cases.

**Summary Of The Paper:**

In this manuscript, time-series forecasting has been studied in an online scenario. For online training of deep neural predictors one has to take into account both the new coming knowledge as well as retaining the learned patterns in the past (the so called stability-plasticity dilemma). The authors reportedly are inspired from the Complementary Learning Systems (CLS) theory,  a fast-and-slow neuroscience learning framework in human. Namely, for information retention, they have used an associative memory and for fast learning  they utilized an adapter per layer, in a Temporal Convolutional Network (TCN).


**Summary Of The Review:**

Although the studied topic is interesting, the current version needs more work to get into the shape that it could be publish in this conference.

Update:  in my opinion, after authors modifications in this phase the manuscript is improved.

---

> ### Author Response · Authors · 2022-11-18
> **Response to Reviewer r44V (part 1)**
>
> **Concern #1: “Many aspect has been explored (Pham et al., 2021a; (Pham et al., 2021b, Arani et al., 2021). Basically, a continual learning scenario has been translated to an online learning scenario”**
>
> We respectfully disagree with the Reviewer that many aspects of our work have been studied in the past. We would like to clarify our two major contributions of: (i) a novel formulation to approach online time series forecasting; and (ii) a novel and non-trivial method inspired from recent continual learning ideas.
>
> First, we emphasize that our work does not simply “translate a continual learning into an online learning scenario”. We study the traditional online time series forecasting problem and propose to approach it from a continual learning point of view. Even so, there are fundamental differences that make directly applying task-free continual learning solutions to online time series forecasting non-trivial, as we discussed in Section 3.1 and empirically verified in the experiments (Table 1). In particular, most of the task-free continual learning frameworks are designed for image classification tasks, which are vastly different from time series data. Particularly, the input (image) and output (label) spaces of images are vastly different: the label space is discrete and the label changes significantly across different tasks. In contrast, time series forecasting demonstrates unique and completely different characteristics: input and output spaces share the same real-valued space, data distributions change gradually over time with no clear task boundaries, and there exists a temporal consistency across consecutive samples, etc.
> Based on these observations, we propose a novel solution: leverage past knowledge to always improve learning in the future, which is akin to forward transfer in continual learning (Section 3.2.1); and remember repeating patterns, which is akin to preventing catastrophic forgetting (Section 3.2.2). We firmly believe our formulation provides a novel perspective to approach online time series forecasting, which is also supported by Reviewers dWpT and MVEA.
>
> Second, fast-and-slow learning is a general idea to continual learning and can motivate different approaches. For example, Pham et al., 2021b proposed to implement fast-and-slow learning as two different objectives (self-supervised (SSL) and supervised learning) while Arani et al., 2021 implemented different learning frequencies for the fast and slow learners. However, they are mainly developed for image data and it is not trivial to extend to time series. For example, Arani et al., 2021 was not originally developed for data streams and required storing multiple copies of the models, resulting in an unfair comparison. Pham et al., 2021b required a good SSL method for representation learning, which is still an open research topic for time series. Therefore, these solutions are not suitable for time series, which motivated us to derive our solution from the ground up to explicitly address time series challenges.
>
> **Concern #2: “Improved the writing of Section 4”**
>
> Thank you for pointing out the unclear notations in the proposed method, which should be Section 3. In light of your comments, we have extensively revised the writing of Section 3 to improve the clarity of FSNet. Particularly, in the beginning of Section 3.2, we provided a summary of the notations used to describe FSNet. Furthermore, we also reminded the definition of each variable when they first appeared in the subsequent equations. We revised Section 3.2.1 to outline the key idea of fast learning in FSNet clearly and describe its mechanism in detail. We hope this revision improves the clarity of our method. Your feedback to this new version will be much appreciated.

---

> > ### Author Response · Authors · 2022-11-18
> > **Response to Reviewer r44V (part 2)**
> >
> > **Concern #3: “Include the exact inference time”**
> >
> > We provided the throughput (samples/second) of different methods on the ETTh2 (H=48) dataset in Table 6, Appendix E.2, also in the following table.
> >
> > | Throughput | ETTh2 | ETTm1 | WTH | ECL | Traffic | S-A |
> > |---|---|---|---|---|---|---|
> > | ER | 46 | 46 | 43 | 42 | 39 | 46 |
> > | DER++ | 45 | 45 | 43 | 42 | 38 | 46 |
> > | TFCL | 29 | 28 | 27 | 27 | 26 | 27 |
> > | MIR | 22 | 22 | 21 | 21 | 30 | 23 |
> > | FSNet | 28 | 28 | 28 | 27 | 27 | 29 |
> >
> > We would like to remind the Reviewer that "learning fast" in our context does not refer to fast running time, but the sample efficiency: FSNet requires less samples to achieve low error rates, especially when a distribution shift happens (as discussed in Section 1 and 2.1). This result is corroborated in Figure 2, where FSNet showed lower error peaks compared to other baselines.
> > Regarding the running time, the Table showed that FSNet is quite efficient compared to common baselines like MIR and TFCL. Nevertheless, ER and DER++ still have a faster running time due to their simplicity.
> >
> > **Concern #4: “Memory overhead analysis”**
> >
> > We provided a memory analysis in Appendix E.2, including a detailed asymptotic analysis and the exact number of parameters.
> >
> > From Table 5 (Appendix E.2), FSNet enjoys better asymptotic complexity compared to replay methods like ER, DER++, because its total complexity does not scale with the size of the look-back window (FSNet does not store the original data).
> >
> > In terms of the number of parameters, we implemented a fair comparison by configuring each method to use the same budget of total parameters, which includes the model, additional training components (adapter and associative memory), and the episodic memory to store data (as we stated in the main paper, Section 4.1, Implementation Details). Table 4 (Appendix E.2) reports the number of parameters of different methods on the ETTh2 data set with H=24. We can see that the total number of parameters are roughly the same for each method.

---

> > ### Comment · Reviewer_r44V · 2022-11-20
> > **marginally above the acceptance threshold**
> >
> > Thank you very much for your response and your attention to the questions.
> > After reading the author's response, I would like to increase my vote for this paper to "marginally above the acceptance threshold". I agree that the presented setup in the context of time series is novel. However, I still have some doubts! How would you argue that recording the presented coefficients in the memory is a better option than simply saving their related gradient vectors ? Specifically, why did you use this form of the coefficients to record in the memory? Finally, about the backbone that you use, TCN, it would be nice to talk more about it and say why you chose it and whether you did any experimentation with other backbones.

---

> > > ### Author Response · Authors · 2022-11-21
> > > **Follow-up response to Reviewer r44V (part 1)**
> > >
> > > We are grateful for your prompt response and for increasing your vote to "marginally above the acceptance threshold." We appreciate your feedback which has further improved our work. Please find our response to your follow-up comments below.
> > >
> > > **Comment #1: “recording the presented coefficients in the memory is a better option than simply saving their related gradient vectors.  Why did you use this form of the coefficients to record in the memory “**
> > >
> > > Thank you for the interesting observation. We choose to store the transformation coefficients (the adapters’ output) instead of other alternatives, such as the gradient EMA (the adapters’ input), because of its **efficiency**. The adapters take the backbone gradient EMA as input and map it to smaller transformation coefficients applied channel-wise on the backbone’s parameters or feature maps. Therefore, these coefficients require much less memory than the gradient EMAs; each has an equal size as the backbone. For example, as reported in Table 4, Appendix E.2, with a memory size M=32, the total associative memory size is roughly equal to a single copy of the backbone network in terms of floating point numbers. On the other hand, storing 32 gradient EMA is equivalent to 32 copies of the backbone and could easily run out of memory on most modern GPUs.
> > >
> > > **Moreover, storing the coefficients allows us to take advantage of both old and new experiences.** Compared to other continual learning benchmarks (e.g., on image data), a repeating pattern in time series may not reappear exactly in the future. For example, electricity consumption this year and last year are likely to have a similar pattern (come from the same distribution), but they may share different values. As a result, we would like to continue learning these patterns from the last time they appeared, given the new knowledge we have accumulated so far.
> > > This approach is implemented by our design: when the associative memory is triggered, FSNet blends the past transformation coefficients with the current one for the current input. As a result, the blended coefficients can take advantage of the new knowledge from the recent time steps and the past knowledge in dealing with recurring patterns. This motivation is also discussed in Sections 3.1 and 3.2.2.
> > >
> > > In summary, storing the adapter’s transformation coefficients is a more effective and efficient way to deal with recurring patterns in online time series forecasting.

---

> > > > ### Author Response · Authors · 2022-11-21
> > > > **Follow-up response to Reviewer r44V (part 2)**
> > > >
> > > > **Comment #2: “The use of TCN backbone”**
> > > >
> > > > Our work focuses on online time series forecasting with deep neural networks. Among the existing architectures, we choose TCN because it is a deep network with a simple design and has shown competitive performances across a wide range of time series applications [a,b].
> > > >
> > > > Although our main paper only reports the results of the standard TCN, FSNet can be easily extended to other feed-forward architectures. We also conducted an experiment comparing among different learning strategies using a deep MLP backbone with ten layers. Except for the deep neural network feature extractor, this experiment follows the same setting as Table 1 in our main paper. Please find the results in the following Table.
> > > >
> > > > | Method  |    | FSNet     |           | DER++     |           | ER    |       | Online TCN |       |
> > > > |---------|----|-----------|-----------|-----------|-----------|-------|-------|------------|-------|
> > > > |         | H  | MSE       | MAE       | MSE       | MAE       | MSE   | MAE   | MSE        | MAE   |
> > > > | ETTh2   | 1  | **0.764** | **0.419** | 0.780     | 0.443     | 0.816 | 0.448 | 0.880      | 0.463 |
> > > > |         | 24 | **0.657** | **0.483** | 0.766     | 0.542     | 0.811 | 0.564 | 1.143      | 0.672 |
> > > > |         | 48 | **0.733** | **0.503** | 0.999     | 0.562     | 1.091 | 0.572 | 1.129      | 0.590 |
> > > > | ETTm1   | 1  | **0.095** | **0.209** | 0.108     | 0.232     | 0.112 | 0.237 | 0.140      | 0.274 |
> > > > |         | 24 | **0.111** | **0.242** | 0.201     | 0.330     | 0.200 | 0.330 | 0.247      | 0.374 |
> > > > |         | 48 | **0.117** | **0.251** | 0.192     | 0.327     | 0.197 | 0.331 | 0.275      | 0.396 |
> > > > | ECL     | 1  | **3.517** | **0.702** | 3.938     | 0.632     | 4.221 | 0.711 | 5.093      | 0.869 |
> > > > |         | 24 | **6.243** | **1.072** | 9.039     | 1.239     | 9.451 | 1.303 | 10.714     | 1.447 |
> > > > |         | 48 | **7.248** | **1.112** | 9.460     | 1.265     | 9.734 | 1.322 | 11.300     | 1.478 |
> > > > | Traffic | 1  | **0.480** | **0.400** | 0.493     | 0.411     | 0.558 | 0.450 | 1.220      | 0.574 |
> > > > |         | 24 | **0.610** | **0.462** | 0.612     | 0.464     | 0.680 | 0.503 | 0.801      | 0.563 |
> > > > | WTH     | 1  | **0.203** | **0.261** | 0.234     | 0.296     | 0.248 | 0.310 | 0.280      | 0.340 |
> > > > |         | 24 | 0.311     | 0.380     | **0.303** | **0.357** | 0.304 | 0.360 | 0.366      | 0.410 |
> > > > |         | 48 | **0.208** | **0.284** | 0.299     | 0.359     | 0.307 | 0.364 | 0.398      | 0.431 |
> > > > | S-A     | 1  | **2.880** | **1.263** | 3.002     | 1.319     | 2.933 | 1.329 | 3.191      | 1.386 |
> > > > |         | 24 | **2.909** | **1.289** | 3.539     | 1.436     | 3.627 | 1.451 | 3.754      | 1.472 |
> > > > | S-G     | 1  | **2.722** | **1.299** | 2.756     | 1.300     | 2.933 | 1.329 | 3.191      | 1.386 |
> > > > |         | 24 | **2.909** | **1.289** | 3.539     | 1.436     | 3.627 | 1.451 | 3.754      | 1.472 |
> > > >
> > > > The result shows a similar conclusion as in our main paper: in most cases, FSNet achieves lower cumulative errors compared to other competitors, often by a significant margin.
> > > > In summary, FSNet is general and can be easily applied to other feedforward architectures with similar results. We chose TCN in our main draft because of its simplicity and competitive performances.
> > > >
> > > > **Reminder to change your recommendation in the original review**
> > > >
> > > > We would like to thank the Reviewer again for willing to change your assessment of our work. However, it seems like the original review still shows your recommendation to be “reject, not good enough”. We would be grateful if the Reviewer could check and change the recommendation accordingly.  Lastly, please let us know if you still have any concerns so that we can address them timely.
> > > >
> > > >
> > > > [a] Yue, Zhihan, et al. "Ts2vec: Towards universal representation of time series." Proceedings of the AAAI Conference on Artificial Intelligence. Vol. 36. No. 8. 2022.
> > > >
> > > > [b] Yan, Jining, et al. "Temporal convolutional networks for the advance prediction of ENSO." Scientific reports 10.1 (2020): 1-15.

---

> > > > ### Comment · Reviewer_r44V · 2022-11-22
> > > > **Thank you for the clarifications**
> > > >
> > > > Thank you  very much for the clarifications, I have already updated my vote.

---

> > > > > ### Author Response · Authors · 2022-11-25
> > > > > **Thank you for the feedback**
> > > > >
> > > > > Thank you again for your valuable comments. We really appreciate your efforts in reviewing our work and updating your score.

---

### Official Review · Reviewer_dWpT · 2022-11-03

**Confidence:** 1
**Correctness:** 3
**Technical Novelty And Significance:** 3
**Empirical Novelty And Significance:** 2
**Recommendation:** 6

**Clarity, Quality, Novelty And Reproducibility:**

However the paper is almost well written, but it seems that understanding the paper is difficult for the reader. Probably by adding more explanation in the introduction and revising the essay, this problem can be solved.

For understanding the video, there are a lot of papers that try to learn from the stream data, such as videos. For example, the slow-fast method. Can we use such methods for time series? If so, why did you not compare your method against those methods



**Strength And Weaknesses:**

The idea is interesting and novel in forecasting the time series

Experimental shows the good performance of proposed method

Authors formulated the problem as a continual task where can be considered a novel approch.

**Summary Of The Paper:**

This paper proposes an efficient learning method (slow and fast) for forecasting the time series. The task formulated as a task free online continual problem. To do so, they introduce the FSNET network where each layer equipped with a memory and an adaptor to adapt with the recent chsnges.

**Summary Of The Review:**

The novelty of paper is marginal, and also the paper should be revised to be easy to read. Also, comparing with previous method is required.

---

> ### Author Response · Authors · 2022-11-18
> **Response to Reviewer dWpT**
>
> **Concern #1: “Adding more explanation in the introduction” and “revising the essay”**
>
> Your feedback is much appreciated. It would be a great help if you could point out the unclear aspect of our writing so that we can further improve it.
>
> For the current revision, we  have tried our best to improve the writing of Section 1 and 3. Particularly:
>
> * We revised the third and fourth paragraph in Section 1 to highlight the main contributions of our work,
>
> * We revised the last two paragraph of Section 3.1 to discuss key differences in our formulation compared to other studies,
>
> * We revised Section 3.2 to carefully detail the fast learning mechanism in FSNet.
>
> We hope that the revised version would be easier to follow and we appreciate your feedback on this newer version.
>
> **Concern #2: comparison with video understanding methods**
>
> Thank you for the suggestion. To the best of our knowledge, **representation learning of each image is critical for video understanding** since each image contains rich semantic information and is a building block for videos. On the other hand, we follow the conventional time series forecasting benchmarks with datasets from various domains such as weather, traffic, which have different characteristics from video data. In time series, **each data point alone provides little information** and time series analysis mostly focuses on modeling the temporal information over time. As a result, it is not trivial to use video understanding methods for forecasting time series, and vice versa.
>
> Next, we would like to clarify if the SlowFast method referred to by the Reviewer was [a]. We understand that SlowFast [a] was proposed to **alleviate the expensive computational cost of analyzing videos**. In particular, SlowFast deploys a fast (small) network with a high frame rate to obtain image representations while using a slow (big) network with a lower frame rate to model the video representation. However, this design is already encapsulated in the TCN backbone used in our study because of its dilated convolution design: shallow layers use small dilation rates to learn from high resolutions while deeper layers use large dilation rates to learn from lower resolutions. Therefore, although FSNet and SlowFast share similar terminologies, they are developed to address completely different challenges in different domains.
>
> For comparison, we conducted a **preliminary** experiment by implementing SlowFast to the TCN backbone used in our work and comparing with other baselines. The Table below reports this result. As we discussed, since SlowFast's properties are already encapsulated by TCN, introducing another network component does not bring significant benefits. In some cases, ETTh2 and ECL, SlowFast even performs slightly worse than using TCN alone.
>
> | Method | ETTh2 |  | ETTm1 |  | ECL |  |
> |---|---|---|---|---|---|---|
> |  | MSE | MAE | MSE | MAE | MSE | MAE |
> | Online TCN | 0.502 | 0.436 | 0.214 | 0.369| 3.309 | 0.635 |
> | SlowFast | 0.517 | 0.432 | 0.132 | 0.259 | 4.598 | 0.911 |
> | FSNet | **0.466** | **0.368** | **0.085** | **0.191** | **3.143** | **0.472** |
>
> In summary, our work focuses on the time series domain. The benchmarks we used included data from various applications that are common for evaluating time series forecasting models. Extending FSNet to the video domain, although interesting, is beyond the scope of our study and we will leave it for future work.
>
> [a] Feichtenhofer, Christoph, et al. "Slowfast networks for video recognition." Proceedings of the IEEE/CVF international conference on computer vision. 2019.

---

> > ### Comment · Reviewer_dWpT · 2022-11-22
> > **New results**
> >
> > Thank you for addressing my concerns and also providing new results in this step.  However, I think the novelty of this work still is an issue.

---

> > > ### Author Response · Authors · 2022-11-22
> > > **Follow-up response to Reviewer dWpT**
> > >
> > > Thank you for the prompt response. We would like to seek your clarification regarding the limited novelty aspects of our work.
> > >
> > > So far, we have clarified and summarized our main contributions to online time series forecasting and continual learning (see our **General response to all Reviewers**). We further reiterated our technical novelty in our response to **Concern #1 of Reviewer r44V**. Our contributions are acknowledged by Reviewers r44V and MVEA. We also clarified the fundamental differences between video understanding and time series forecasting in our response to your original comment (**Concern #2 of Reviewer dWpT**). Lastly, we clarified the possible confusion in terminologies usages between our work and SlowFast [a], and empirically compared with SlowFast to support our arguments.
> > >
> > > Therefore, we would be grateful if Reviewer dWpT could elaborate more on the limited novelty aspects of our work so that we can further address them.
> > >
> > > [a] Feichtenhofer, Christoph, et al. "Slowfast networks for video recognition." Proceedings of the IEEE/CVF international conference on computer vision. 2019.

---

> ### Author Response · Authors · 2022-11-25
> **Further discussions**
>
> Dear Reviewer dWpT,
>
> Thank you again for the valuable comments and feedback. We hope that you have had time go through our revised manuscript and our clarifications to your concerns. We would like to check with you if our responses have been satisfactory. We look forward to further discuss with you if you have any additional comments.
>
> Best regards,
> Authors

---

> ### Author Response · Authors · 2022-12-05
> **Gentle Reminder**
>
> Dear Reviewer dWpT,
>
> Thank you again for your time and constructive comments. As the deadline for discussion is approaching, we look forward to seeing whether our responses properly address your concerns.
>
> We have done our utmost to address your concerns.
>
> *  We extensively revised the writing to make our manuscript easier to follow.
>
> * We clarified the fundamental differences between our work and the video understanding literature, which is accompanied by an experiment to support our claim.
>
> We hope the Reviewer had time to review our revised manuscript and responses. We would be grateful if the Reviewer could elaborate more on the potential limitations of our work so that we can address them timely.
>
> Best Regards,
>
> Authors

---

### Author Response · Authors · 2022-11-18
**General response to all Reviewers**

We thank the Reviewers for the insightful comments and detailed feedback. We were delighted when Reviewers dWpT, r44V, and MVEA agreed that **our idea is interesting and critical in many practical cases**. Reviewers dWpT and MVEA found **our paper to be mostly well-written** and **the technical details easy to follow**. The Reviewers found **our proposed method to be intuitive, interesting** (Reviewer MVEA), **with good performances over a number of baselines** (Reviewer dWpT and MVEA).

There is a shared concern from Reviewers dWpT and r44V **regarding the novelty of our work** in comparison to existing studies in the video understanding literature and fast-and-slow learning. We would like to emphasize that **our work focuses on time series forecasting** and addresses the challenge of learning fast by dealing with distribution shifts with deep neural networks. Our key contribution is a novel solution to online time series forecasting inspired by continual learning. It is important to note that **it is not trivial to directly apply common continual learning frameworks on image data to time series** due to the differences between the two types of data such as discrete vs continuous label, distinct label differences among tasks, etc. (as we discussed in Section 3.1). Therefore, we derive FSNet **from the ground up** by proposing a fast learning mechanism that was not explored in the literature: using adapter to capture temporal smoothness in time series, and an adapter-memory interaction to remember repeating patterns.
We firmly believe that our work contributes substantially to both continual learning and time series forecasting literatures by showing advances in continual learning can inspire solutions to real-world problems.
We refer the Reviewers to our our response to **Concern #2 of Reviewer dWpT** for a comparison with video understanding methods; and our response to **Concern #1 of Reviewer r44V** for a detailed discussion regarding the novelty of our work.

In the revision, we took into account the Reviewers’ comments and substantially improved our manuscript. First, we extensively revised Section 1 and 3 to better present FSNet’s idea, clarify its technical details, and emphasize its contributions over existing studies. We added pointers tot various results in the Appendix to address Reviewers r44V and MVEA’s concerns, including a visualization of FSNet activation patterns, robustness to hyper-parameters, and complexity analyses. Key changes will be highlighted in blue.

---

### Decision · Program_Chairs · 2023-01-20

**Decision:**

Accept: poster

**Justification For Why Not Higher Score:**

Unless there is a mechanism in the review process that allows us to review the revised paper before publication, I don’t feel comfortable of recommending for spotlight acceptance.


**Justification For Why Not Lower Score:**

This paper was a borderline paper based on the initial reviews. After much deliberation with the authors (involving all except one reviewer who has low confidence of his/her recommendation) during the discussion period, all reviewers are now supportive of accepting the paper although the support is not strong (all 6). The authors are recommended to consider the comments and suggestions in the reviews and the subsequent discussions to revise their paper.


**Metareview: Summary, Strengths And Weaknesses:**

We thank the reviewers for responding to our comments and elaborating on a number of points which are not well articulated in the paper. We recognize that the topic studied in this paper are relevant to practical situations and that the problem formulation is novel in the context of time series forecasting. The proposed method also shows good performance in the experiments. Nevertheless, a number of comments have been raised in our reviews. Some of them might be related to presentation. For this work to appeal to a wider audience, the authors are highly recommended to take the comments and suggestions of the reviewers seriously in revising their paper before publication. Doing so may help the paper achieve the impact that it deserves.


**Note From Pc:**

if the above contains the word "oral" or "spotlight" please see: "oral" presentation means -> notable-top-5% and "spotlight" means -> notable-top-25%. As stated in our emails, we are disassociating presentation type from AC recommendations